# BASE-Q: Bias and Asymmetric Scaling Enhanced Rotational Quantization for Large Language Models

## Abstract

Rotation-based methods have become essential for state-of-the-art LLM quantization by effectively mitigating outliers in weights and activations. Current approaches predominantly focus on optimizing the global rotation matrix to achieve marginal accuracy improvements—a strategy that incurs prohibitive computational costs through full-model backpropagation while offering limited practical utility. We fundamentally reassess this optimization paradigm and identify two critical error sources that persist even under optimal rotation conditions: (i) channel mean misalignment, which amplifies rounding errors during quantization, and (ii) clipping-induced energy loss, which is exacerbated by the rotation-induced Gaussian-like distributions. Our analysis reveals that directly addressing these issues offers a more effective path to achieving high quantization accuracy. Based on these insights, we introduce **BASE-Q**, a lightweight quantization framework that circumvents expensive global rotation learning. **BASE-Q** employs simple yet powerful transformer-block-wise correction strategy: **bias correction** to eliminate channel mean variance and **asymmetric scaling** to compensate for clipping-induced energy loss. This blockwise strategy drastically reduces optimization overhead, enabling efficient quantization of 70B parameter models on a single GPU. Extensive experiments across diverse LLMs and benchmarks validate the effectiveness of BASE-Q, narrowing the accuracy gap to full-precision models by 50.5%, 42.9%, and 29.2% compared to previous rotation method QuaRot, SpinQuant, and OSTQuant respectively, demonstrating the superiority of our lightweight paradigm.

## 1 Introduction

Large language models (LLMs) (Touvron et al., 2023; Bai & et al., 2023; Dubey & et al., 2024; Touvron & et al., 2023; DeepSeek-AI, 2024) drive advances across diverse natural language tasks, but their ever-increasing scales present significant challenges for efficient deployment, particularly regarding inference latency and memory consumption. Low-precision quantization (Nagel et al., 2020; Li et al., 2021; Liu et al., 2022; 2025) (e.g., 4 bits or lower) represents a crucial technique for addressing these computational bottlenecks. However, aggressively reducing numerical precision to such low bit-widths results in substantial accuracy degradation. The fundamental challenge underlying this degradation stems from the presence of significant outliers in both activations and weights (Wei et al., 2023; Dettmers et al., 2022), which necessitate wider quantization ranges that consequently amplify quantization errors.

To mitigate the adverse effects of outliers in LLM quantization, recent approaches have leveraged linear equivalent transformations, particularly scaling and rotation operations. SmoothQuant (Xiao et al., 2022) pioneered this direction by introducing channel-wise scaling to reduce activation variance, effectively smoothing the distribution and achieving robust improvements in INT8 post-training quantization (PTQ) across various LLMs. Building on this direction, QuaRot (Ashkboos et al., 2024) introduces Hadamard rotations as another form of equivalent transformation to redistribute outliers and reduce channel-wise disparities, enabling robust INT4 PTQ performance. Since RMSNorm in LLMs normalizes activation energy across residual blocks while preserving this en-

ergy under rotations, identical rotation matrix can be shared across all transformer blocks and fused into adjacent linear layers without incurring additional computation overhead.

Building upon the effectiveness of fixed rotations, a prevailing trend in subsequent research (e.g., SpinQuant (Liu et al., 2024), OSTQuant (Hu et al., 2025)) has pursued further accuracy improvements by learning a single, globally shared rotation matrix. We argue that this paradigm, while seemingly intuitive, is fundamentally flawed for two critical reasons. Firstly, it incurs prohibitive optimization costs due to full-model backpropagation, rendering it impractical for truly large-scale models. More importantly, we find that the core assumption—that an 'optimal' global rotation is the key to minimizing quantization error—is misguided. A single transformation shared across dozens of diverse transformer blocks inherently lacks the expressiveness to simultaneously suppress outliers and accommodate block-specific distributional variations. This theoretical limitation explains the observed marginal improvements and necessitates a paradigm shift away from global optimization.

Motivated by this insight, we shift the focus from optimizing the rotation itself to directly correcting the residual errors it leaves behind. Our analysis identifies two dominant, yet previously overlooked, error sources: **(i) Rounding Error from Channel Mean Misalignment**: We reveal that significant variance persists among channel-wise means even after rotation, as a single global rotation cannot simultaneously nullify this variance across all layers. We address this through a lightweight, block-specific bias correction that precisely realigns channel means with negligible computational overhead. **(ii) Clipping Error from Distributional Shift**: We uncover that rotation, by rendering activations more Gaussian-like, paradoxically increases the energy loss from quantization clipping, thereby violating the theoretical equivalence that rotation aims to preserve. We counteract this through a block-aware asymmetric scaling strategy that restores signal energy post-clipping. By circumventing the costly and ineffective pursuit of a perfect global rotation, our framework, BASE-Q, employs these simple yet targeted corrections at the block level, achieving superior accuracy while enabling efficient and memory-friendly optimization.

Our main contributions are:

- **Theoretical Insight**. We provide a systematic analysis revealing key components of quantization error after rotation, elucidating why existing approaches plateau and identifying opportunities for further optimization.

- **Efficient Transformer-Block-wise Correction**. We propose BASE-Q, a novel quantization framework that employs blockwise channel bias correction and asymmetric scaling under fixed rotations, achieving significant performance enhancement without the expensive full-model optimization.

- **Extensive Empirical Validation**. Extensive experiments across diverse LLMs and benchmarks validate the effectiveness of BASE-Q, narrowing the accuracy gap to full-precision models by 50.5%, 42.9%, and 29.2% compared to previous rotation method QuaRot (Ashkboos et al., 2024), SpinQuant (Liu et al., 2024), and OSTQuant (Hu et al., 2025) respectively.

## 2 RELATED WORK

**Equivalent Transformations in LLM Quantization.** Post-training quantization (PTQ) has attracted considerable attention for the efficient deployment of LLMs, yet remains challenging due to frequent outliers in both activations and weights. AWQ (Lin et al., 2024b) introduced channel-wise scaling for weight-only PTQ, while SmoothQuant (Xiao et al., 2022) employed rescaling for activations and weights to suppress outlier effects and enable robust INT8 quantization. OmniQuant (Shao et al., 2023) extended this concept by introducing learnable scaling coefficients for each submodule, allowing for finer-grained adaptation across network components. AffineQuant (Ma et al., 2024) further generalized these ideas by employing learnable affine transformations to jointly align mean and variance before quantization. Beyond scaling-based methods, QuIP (Chee et al., 2023; Tseng et al., 2024) first applied rotation transformations for weight-only PTQ. QuaRot (Ashkboos et al., 2024) proposed applying Hadamard rotations to both activations and weights, rendering distributions more Gaussian and further suppressing outliers, thus simplifying the quantization process. DuQuant (Lin et al., 2024a) employed rotation and permutation to more effectively eliminate out-

liers. DFRot (Xiang & Zhang, 2024) attributes the success of Hadamard transforms to their handling of rare tokens with massive activations and proposes a weighted loss function combined with an alternating Procrustes-based optimization to learn a globally improved rotation matrix. SpinQuant (Liu et al., 2024) advanced this direction by learning optimal rotation matrices from calibration data, achieving lower quantization errors at the cost of greater computational and memory requirements. OSTQuant (Hu et al., 2025) unified learnable rotations and scaling within a single framework, providing additional flexibility and consistently outperforming previous methods on various LLM benchmarks. FlatQuant (Sun et al., 2024) employed layer-wise learned online matrix transforms to improve quantized linear layers, although at the cost of increased inference overhead and parameter count.

## 3 ERROR ANALYSIS OF ROTATION QUANTIZATION

In this section, we theoretically analyze the errors derived by rotation-based quantization, specifically focusing on the Hadamard transformation. We begin by investigating the error reduction mechanism inherent in this transformation, deriving quantitative expressions for both rounding and clipping errors in Section 3.1. Subsequently, in Sections 3.2 and 3.3, we provide an in-depth analysis of these error sources, uncovering potential components that are amenable to further optimization and providing key insights for methodological improvements. Finally, drawing inspiration from OSTQuant (Hu et al., 2025), which demonstrates complementary effects of scaling and rotation, we interpret and substantiate this synergy through our error decomposition framework in Section 3.4.

**Notation**. Throughout this paper, we follow a consistent notational convention. We denote matrices by uppercase bold letters (e.g., $\boldsymbol{W}$), vectors by lowercase bold letters (e.g., $\boldsymbol{x}$), and scalars by regular lowercase letters (e.g., $s$). All vectors are considered column vectors unless specified otherwise.

### 3.1 QUANTIZATION ERROR ANALYSIS FOR HADAMARD ROTATIONS

The key challenge in LLM quantization is the presence of extreme outliers in the activation channels, which significantly increases the dynamic range and severely impedes quantization performance. To elucidate how Hadamard rotations mitigate this issue, consider a token activation $\boldsymbol{x} \in \mathbb{R}^n$ with outlier values structured as follows:

$$\boldsymbol{x} = \boldsymbol{g} + \sum_{i \in \mathcal{O}} a_i \delta \boldsymbol{e}_i, \tag{1}$$

where $\boldsymbol{g} \sim \mathcal{N}(\mu, \delta^2 \boldsymbol{I})$ represents the 'main mass' as a Gaussian component, and each outlier channel $i \in \mathcal{O}$ is modeled as a one-hot vector $\boldsymbol{e}_i$ of amplitude $a_i \delta$ with $a_i \gg 1$ and $|\mathcal{O}| \ll n$. When an orthogonal Hadamard matrix $\boldsymbol{H}$ is applied (where each entry is $\pm n^{-\frac{1}{2}}$), the transformed activation becomes:

$$\boldsymbol{H}\boldsymbol{x} = \boldsymbol{H}\boldsymbol{g} + \sum_{i \in \mathcal{O}} a_i \delta \boldsymbol{H}\boldsymbol{e}_i. \tag{2}$$

Since Gaussian distributions are invariant under orthogonal transformations, the main mass $\boldsymbol{H}\boldsymbol{g}$ remains Gaussian. Critically, each outlier term $a_i \delta \boldsymbol{e}_i$ is now assigned to the $i$-th column of $\boldsymbol{H}$, a dense vector whose large amplitude $a_i \delta$ is evenly distributed across all channels, with each channel receiving only $a_i \delta / \sqrt{n}$. For large $n$, the per-channel impact of any outlier is significantly diminished effectively, outliers are 'absorbed' into the Gaussian bulk, yielding a distribution that is far more conducive to quantization.

This mechanism extends beyond simple Gaussian activations to more general cases where activations exhibit correlated structures or multiple modes (e.g., Gaussian mixtures or channel-wise mean/variance discrepancies). In such cases, applying a principal component transformation to diagonalize the covariance matrix, followed by a Hadamard rotation, further balances the marginal variances and minimizes the prominence of outliers:

$$\boldsymbol{x}' = \boldsymbol{H}\boldsymbol{U}^T(\boldsymbol{x} - \mathbb{E}[\boldsymbol{x}]), \tag{3}$$

where $\boldsymbol{U}$ is the matrix whose columns are the principal component vectors. To quantify the effect on quantization, consider uniform quantization with per-channel clipping:

$$\boldsymbol{x}_q = F_{clip}(F_{round}(\frac{\boldsymbol{x} - z}{s}), 0, 2^b - 1) * s + z, \tag{4}$$

where $z$ and $s$ denote the quantization lower bound and step size, respectively, and $b$ represents the target bit-width. The overall quantization error comprises two components:

- **Rounding error**, with expected $\ell_2$ energy per channel: $\mathbb{E}[\|\varepsilon_{\text{round}}\|_2^2] = \frac{\Delta^2}{12}$, where $\Delta$ denotes the quantization step size (bin width). This classical result stems from modeling the rounding error as uniform noise, a concept we will elaborate on in Section 3.4 (see Eq. 10).

- **Clipping error**, defined by the mass outside quantization bounds: $\mathbb{E}[\|\varepsilon_{\text{clip}}\|_2^2] = \int_{-\infty}^{z} x^2 P(x)dx + \int_{z+\Delta}^{+\infty} x^2 P(x)dx$, where $\Delta = s(2^b-1)$ and $P(x)$ is the empirical channel distribution.

By dispersing outlier energy and compressing the activation dynamic range, Hadamard transformations significantly reduce rounding errors by enabling smaller quantization steps $s$.

## 3.2 ROUNDING ERRORS FROM MISALIGNED CHANNEL MEANS

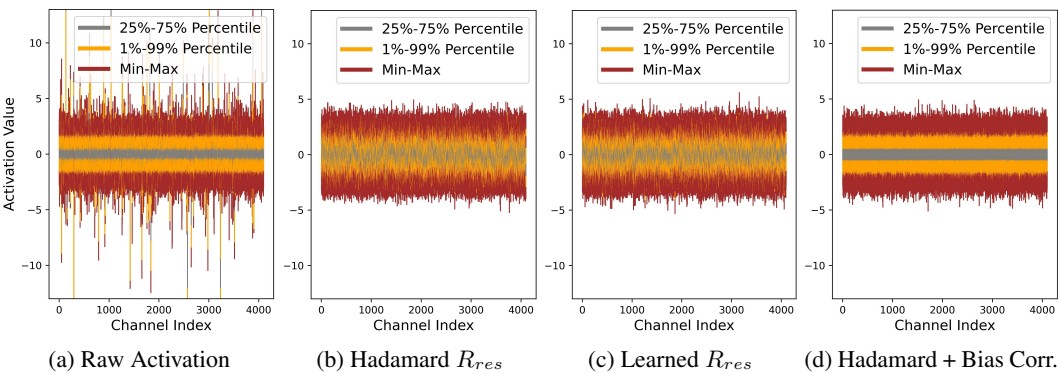

| (a) Raw Activation | (b) Hadamard $R_{res}$ | (c) Learned $R_{res}$ | (d) Hadamard + Bias Corr. |

Figure 1: (a) Raw activation distribution of the first MLP block in Llama3-8B. (b) Hadamard rotation suppresses outliers but leaves residual inter-channel misalignment. (c) Learned rotations also fail to address this misalignment. (d) Our bias correction eliminates most inter-channel mean variability, thereby reducing $Var(\mu_j)$, which is the main rounding error component as formalized in Equations (5) and (6). Additional visualizations demonstrating this effect are provided in Section D.

As demonstrated in the previous section, the energy of the rounding error is proportional to the quantization scale, $s = \frac{\Delta}{2^b-1}$. As rotations gaussianize the activation distribution, the quantization range can be considered proportional to the standard deviation of activations, $\sigma$. Therefore, the expected rounding error satisfies:

$$\mathbb{E}[\|\varepsilon_{\text{round}}\|_2^2] \propto s^2 \propto \sigma^2. \tag{5}$$

The total variance $\sigma^2$ of activations across all channels can be formally decomposed using the Law of Total Variance, which states:

$$\sigma^2 = \frac{1}{n}\sum_{j=1}^{n}\sigma_j^2 + Var(\mu_j), \tag{6}$$

where $\sigma_j^2$ and $\mu_j$ denote the variance and mean of the $j$-th channel, respectively. The first term, representing the average channel variance, remains invariant under orthogonal transformations and is determined by the data distribution. Ideally, an optimal rotation would align all channel means, reducing $Var(\mu_j)$ to zero. However, we observe that even globally learned rotations (e.g., in Spin-Quant) fail to achieve this alignment, yielding only marginal improvements over fixed rotations in this regard (see Figure 1(c)). This limitation motivates our direct approach. As demonstrated in Figure 1(d) and Table 5, our blockwise bias correction effectively eliminates $Var(\mu_j)$, which renders the choice of the global rotation matrix $R_{res}$ insignificant. Consequently, models with different $R_{res}$ choices converge to consistently high accuracy. This stands in stark contrast to prior work, where learning $R_{res}$ is crucial to performance stability. Ultimately, our framework fully subsumes

the optimization potential of a learnable global rotation, allowing its computationally expensive optimization to be entirely removed from the quantization pipeline.

Ideally, optimizing rotation should align all channel means to achieve $Var(\mu_j) = 0$. In practice, however, the LLM shares a single rotation $\boldsymbol{R}_{res}$ across all transformer blocks, despite blockwise variation in bias. The limited expressiveness of a global rotation renders it infeasible to simultaneously eliminate outliers and achieve mean alignment. As a result, globally learned rotations (e.g., SpinQuant) can only minimize $Var(\mu_j)$ in a least-loss sense but cannot completely eliminate this error arising from mean discrepancies. This limitation motivates our explicit bias correction strategy, which precisely cancels the variance-of-means term post-rotation. Moreover, this approach supports blockwise optimization, circumventing the computational burden of full-model optimization.

### 3.3 CLIPPING ELIMINATES NON-NEGLIGIBLE ENERGY

Orthogonal rotation, such as the Hadamard transformation, significantly alters the distribution of activations. While raw activations typically exhibit a heavy-tailed distribution with sparse outliers, post-rotation activations approximate a Gaussian distribution. This transformation has critical implications: unlike heavy-tailed distributions where extreme values are rare, the Gaussian shape concentrates a larger proportion of activation energy towards the distribution tails. Consequently, applying a fixed clipping threshold removes a non-negligible amount of energy from values exceeding the threshold.

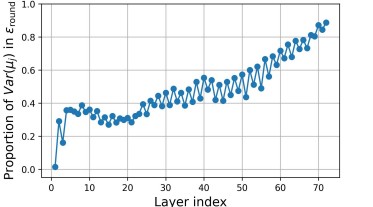
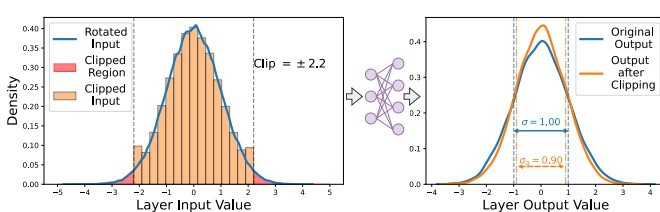

Figure 2: Misalignment causes up to 85% layer-wise rounding error in Qwen2.5-3B.

Figure 3: (Left) MSE-optimal clipping on rotated activations results in a 18.4% loss of layer's input energy. (Right) This induces significant discrepancies in the layer output.

Formally, consider activations $\boldsymbol{x} \sim \mathcal{N}(\mu, \sigma^2)$ quantized to $b = 4$ bits (INT4), the minimum MSE-optimal clipping threshold is commonly determined as:

$$\theta^* = \arg\min_\theta \mathbb{E}[\|\boldsymbol{x} - (F_{clip}(F_{round}(\frac{\boldsymbol{x} - \theta}{2\theta/(2^b - 1)}), 0, 2^b - 1) + \theta)\|_2] \approx 2.2\sigma \qquad (7)$$

Here, $\sigma$ denotes the standard deviation of the post-rotation activations, which are approximately Gaussian. The factor $\theta^* \approx 2.2\sigma$ is the empirically found optimal clipping threshold coefficient for minimizing. At this threshold, we can quantify the proportion of total activation energy contained within the clipped regions (i.e., the energy loss). For a zero-mean Gaussian distribution, the expected $L_2$ energy of the clipped values constitutes approximately 18.4% of the total energy:

$$\mathbb{E}[\|\varepsilon_{\text{clip}}\|_2] = \int_{-\infty}^{-2.2\sigma} x^2 P(x)dx + \int_{+2.2\sigma}^{+\infty} x^2 P(x)dx \approx 18.4\%, \qquad (8)$$

where $P(x)$ denotes the probability density function. 18.4% of the energy is **non-negligible** in the context of deep neural networks. When this clipping is applied layer by layer throughout a deep network, the accumulated energy loss becomes substantial. The empirical consequence of this breakdown is illustrated in Figure 3, where the distribution of the layer output significantly mismatches the original. Moreover, in practical LLM architectures, neither globally nor per-layer optimized clipping bounds can fully resolve this energy loss, as joint optimization across all layers is intractable. This limitation explains the diminishing returns observed in aggressive clipping strategies for rotation-based quantization schemes, even with parameter tuning.

To address this issue, we introduce asymmetric scaling for each activation quantization step, which restores the energy loss and maintains accurate signal magnitude, thereby improving both theoretical fidelity and empirical performance.

### 3.4 Role of scaling in rotational quantization

Combining scaling with orthogonal transformations (rotations) has been shown to be crucial for enhancing quantization performance. To understand this synergy, we analyze the effect of quantization noise from the perspective of the original, pre-rotation space. In the rotated space, the rounding noise introduced by a uniform quantizer can be modeled as additive noise $\epsilon$ where each component is an independent random variable uniformly distributed over the quantization bin:

$$\epsilon \sim \mathcal{U}(-\frac{\Delta}{2}, \frac{\Delta}{2}), \tag{9}$$

where $\Delta$ denote the rounding noise and quantization bin width. The variance of this noise is $\frac{\Delta^2}{12}$. Since orthogonal transform preserves the variance, this noise manifests as additive isotropic Gaussian noise in the original space, which becomes:

$$\epsilon \sim \mathcal{N}(0, \frac{\Delta^2}{12}). \tag{10}$$

Referring to the matrix multiplication of weights and activations in the specific layer, the layer output includes terms arising from error propagation:

$$(w + \epsilon_w) \cdot (a + \epsilon_a) = w \cdot a + (w \cdot \epsilon_a + \epsilon_w \cdot a) + \epsilon_w \cdot \epsilon_a, \tag{11}$$

where $w$ and $a$ represent the original weights and activations respectively, while $\epsilon_a$ and $\epsilon_w$ denote their corresponding quantization errors.

Introducing per-channel symmetric scaling factors $s$ (applied to $w$) and $1/s$ (applied to $a$) allows balancing the propagated error energy between these terms. The total variance of the noisy product, dominated by $w \cdot \epsilon_a$ and $\epsilon_w \cdot a$, achieves its minimum when their variances are equal. By the arithmetic-geometric mean (AM-GM) inequality, the optimal scaling factor satisfies:

$$s^2 = \frac{\mathbb{E}[|\boldsymbol{w}|_2]}{\mathbb{E}[|\boldsymbol{a}|_2]}. \tag{12}$$

This choice of symmetric scaling enables optimal allocation of quantization noise energy. Unfortunately, symmetric scaling breaks this per-token energy, making it incompatible with fusible rotations for layers immediately following an RMSNorm (i.e., $Q$, $K$, $V$, $Up$, $Gate$ projections). It can only be effectively paired with fusible rotation in layers like the $O$ and $Down$ projections.

In summary, our analysis in this section has deconstructed the residual errors in rotational quantization, identifying two key challenges that persist even after outlier smoothing: 1) significant rounding errors stemming from the variance of channel means (Sec. 3.2), and 2) substantial activation energy loss due to clipping (Sec. 3.3). These findings suggest that solely optimizing the rotation matrix offers diminishing returns. To overcome these fundamental limitations, in the following section, we introduce BASE-Q, a framework designed to directly target and correct these two specific error sources with high efficiency.

## 4 BASE-Q

As discussed in Sections 3.2 and 3.3, while rotation-based quantization methods effectively smooth outliers, they exhibit three notable limitations: (a) Channel-mean variance cannot be eliminated through rotation-only optimization, leading to a persistent rounding error term (as formalized in Equations (5) and (6)). (b) Implementing flexible and optimal activation clipping schemes within standard rotation-based PTQ is challenging, resulting in cumulative energy loss across layers. (c) Optimizing full-network rotation parameters requires prohibitive GPU memory and computational resources, especially for large models. To address these issues, we propose **BASE-Q** (**B**ias and **A**symmetric **S**caling **E**nhanced **Q**uantization), a lightweight yet powerful quantization framework that strengthens rotation-based methods through two key innovations: explicit **bias correction** to eliminate channel mean variance, and **asymmetric scaling** to improve clipping and quantizer fit at each block. The blockwise optimization capability circumvents the heavy cost and complexity of full-model rotation learning, thereby achieving efficiency and superior quantization performance.

Figure 4: An overview of **BASE-Q**, highlighting three key design features: (a) channel-wise bias correction to reduce rounding error in activation quantization. (b) asymmetric scaling to compensate for the loss of computational equivalence caused by the clipping; (c) elimination of $\boldsymbol{R}_{res}$ learning, thereby avoiding full model optimization. This mechanism matches the parameter count and hardware fusion efficiency of typical scaling strategies.

**Fundamental Rotation Settings.** Figure 4 provides a schematic of our BASE-Q. The core optimized transformations within each transformer block include the global shared residual rotation ($\boldsymbol{R}_{res}$), the blockwise value rotation ($\boldsymbol{R}_v$), and the online rotations ($\boldsymbol{R}_{qk}$ and $\boldsymbol{R}_{down}$). $\boldsymbol{R}_{res}$ is applied to the residual pathway and can be fused into linear weights without additional inference overhead. Since the residual path connects all blocks, a single shared rotation is used throughout. As our method explicitly addresses channel mean variance and optimizes quantization for activations, we configure $\boldsymbol{R}_{res}$ to primarily optimize weight statistics. Following Section 3.1, we compute the PCA transform $\boldsymbol{U}$ across all related linear weights, and compose it with the Hadamard transformation:

$$\boldsymbol{R}_{res} = \boldsymbol{U}^T \boldsymbol{H}, \tag{13}$$

where $\boldsymbol{U}$ is derived from the covariance of all associated linear weights. $\boldsymbol{R}_v$ can be exactly fused layer-wise into linear parameters and supports independent, block-level optimization. We employ PyTorch's built-in Cayley orthogonal mapping for rotation learning, initialized as a Hadamard matrix. $\boldsymbol{R}_{qk}$ and $\boldsymbol{R}_{down}$ require on-the-fly computation during inference. To reduce memory and computation overhead, we adopt Hadamard rotations, which contain binary elements and can be efficiently implemented through fast algorithms.

**Bias Correction.** To directly address the rounding error component caused by $Var(\mu_j)$, we introduce a learnable bias prior to activation quantization. After quantized inference and the corresponding linear projection, this bias is subtracted, yielding:

$$\boldsymbol{y} = Q_w(\boldsymbol{W}\boldsymbol{s}\boldsymbol{R}) \, Q_a([\boldsymbol{R}^{-1}\boldsymbol{x} - \boldsymbol{b}^c]) + \underbrace{\boldsymbol{W}\boldsymbol{R}\boldsymbol{b}^c + \boldsymbol{b}}_{fused\ bias} \tag{14}$$

where $\boldsymbol{W}\boldsymbol{R}$ denotes rotated weights and $\boldsymbol{b}$ represents the usual layer bias. The learnable bias correction terms—including $\boldsymbol{b}_{qkv}^c$, $\boldsymbol{b}_o^c$, $\boldsymbol{b}_{up}^c$ and $\boldsymbol{b}_{down}^c$-incur negligible parameter overhead (less than 0.1% of original model size) and are highly efficient to optimize.

**Asymmetric Scaling.** To counteract the energy loss from clipping in post-rotation Gaussian-like distributions, we introduce an per-quantizer asymmetric scaling $\boldsymbol{s}^a$. This asymmetric scaling $\boldsymbol{s}^a$ adjusts the quantization range at inference time to better accommodate activation statistics and clipping requirements:

$$Y = Q_w(\boldsymbol{W}\boldsymbol{s}\boldsymbol{R}) \, Q_a(\boldsymbol{s}^a \boldsymbol{R}^{-1} \boldsymbol{s}^{-1} \boldsymbol{x} - \boldsymbol{b}^c) + (\boldsymbol{W}\boldsymbol{R}\boldsymbol{b}^c + \boldsymbol{b}) \tag{15}$$

**Blockwise Optimization.** Combining these techniques, our blockwise optimization strategy jointly optimizes $\boldsymbol{R}_v$, $\boldsymbol{b}_i^c$, $\boldsymbol{s}_j$, $\boldsymbol{s}_i^a$, and activation clipping threshold $\alpha_i$ per block via minimizing the MSE between the floating-point and quantized outputs:

$$\underset{\boldsymbol{R}_v, \boldsymbol{b}_i^c, \boldsymbol{s}_j, \boldsymbol{s}_i^a, \alpha_k}{argmin} \quad \mathcal{L}_{mse}(\boldsymbol{y}_{FP}, \boldsymbol{y}_Q; \boldsymbol{R}_v, \boldsymbol{b}_i^c, \boldsymbol{s}_j, \boldsymbol{s}_i^a, \alpha_i, \theta) \tag{16}$$

where $\boldsymbol{y}_{FP}$ and $\boldsymbol{y}_Q$ denote the reference and quantized outputs respectively, while $\theta$ represents the set of frozen model and rotation parameters. This framework achieves state-of-the-art quantization accuracy with negligible memory and computational overhead during the quantization process.

## 5 EXPERIMENT

**Models and Tasks.** We evaluate our method on 12 open-source LLMs spanning various sizes: Llama-2-7/13/70B (Touvron et al., 2023), Llama-3-8/70B (Dubey & et al., 2024), Llama-3.1-8/70B, Llama-3.2-1/3B, and Qwen2.5-3/14/32B (Yang et al., 2024). We report perplexity on the Wikitext-2 dataset (Merity et al., 2016) and zero-shot accuracy on nine downstream tasks, including ARC-Easy and ARC-Challenge(Clark et al., 2018), BoolQ (Clark et al., 2019), HellaSwag (Zellers et al., 2019), LAMBADA (Radford et al., 2019), OpenBookQA (Mihaylov et al., 2018), PIQA (Bisk et al., 2020), SIQA (Sap et al., 2019), and WinoGrande (Sakaguchi et al., 2021). We compare against baseline methods: QuaRot (Ashkboos et al., 2024), which employs fixed Hadamard rotations; Spin-Quant (Liu et al., 2024), which leverages learned rotations; and OSTQuant (Hu et al., 2025), which incorporates both learned rotations and scaling.

**Deployment Details.** Our quantization framework is implemented in PyTorch and evaluated using lm-eval (Gao et al., 2024). We apply per-token asymmetric dynamic quantization for activations, per-head asymmetric dynamic quantization for KV-cache, and per-channel symmetric quantization for weights. Calibration is performed using 128 samples from Wikitext-2. For each transformer block, we first train symmetric scaling for 3 epochs, then apply weight quantization using GPTQ (Frantar et al., 2022), and finally train bias terms, asymmetric scaling, and learnable clipping factors for 5 epochs. Learning rates are initialized as 1e-2 for scaling and clipping, 1e-3 for bias, with cosine decay applied throughout training. The quantization process requires approximately 0.7 hours for 3B models and 10 hours for 70B models on a single A800 GPU. This highlights the efficiency of our blockwise approach, which avoids the substantial resource requirements of full-model optimization methods like SpinQuant, which necessitates at least 5 A800 GPUs and 30 GPU-hours for a 70B model.

### 5.1 MAIN RESULTS

Table 1 presents the overall performance of BASE-Q and baseline methods under W4A4KV4 quantization, reporting perplexity on Wikitext-2 and the average accuracy across nine zero-shot tasks. Since BASE-Q is explicitly designed to tackle the most challenging 4-bit activation quantization, we focus solely on the W4A4KV4 configuration. Across all evaluated models, BASE-Q consistently achieves superior results, achieving an average accuracy degradation of only 3.40% from full precision. In comparison, the average accuracy drops for QuaRot, SpinQuant, and OSTQuant are 6.87%, 5.95%, and 4.80%, respectively (averaged over eight supported models). This corresponds to BASE-Q reducing the performance gap to full precision by 50.5%, 42.9%, and 29.2% relative to QuaRot, SpinQuant, and OSTQuant, respectively. Notably, BASE-Q performs exceptionally well on Qwen2.5-3B, where all existing methods suffer severe degradation (perplexity increases by more than 100%). These results strongly validate our theoretical insights and demonstrate the practical effectiveness of our proposed bias correction and asymmetric scaling strategies. To provide a more comprehensive evaluation, we additionally test our method on the complex, multi-subject MMLU benchmark in Table 2.

Additional evaluations across different bit-width settings (W6A6KV6, W3A4KV4, W2A4KV4) are presented in Tables 3, 10 and 11, including comparisons against FlatQuant. FlatQuant improves quantization performance via additional learned online transforms for each layer. While this approach proves effective at certain bit-widths, it introduces substantial computational overhead compared to methods based on fusible rotation. Furthermore, we observe that these complex transforms are susceptible to numerical instability. Notably, FlatQuant fails on the Qwen2.5-3B model under our W3A4 setting and struggles to converge under W2A4 across most models, highlighting a practical robustness challenge at ultra-low precision. In contrast, BASE-Q's method of combining a fusible global rotation with lightweight corrections proves to be both more stable and more efficient across all tested bit-widths.

### 5.2 PREFILL ACCELERATION ON GPU

We benchmark INT4 quantization during the compute-bound prefill stage on an NVIDIA 3090 GPU. Int4 matrix multiplications are implemented using NVIDIA's Cutlass library, while other custom operators are written in Triton to ensure flexibility and speed. To minimize computational overhead,

Table 1: Comparison of Wikitext-2 perplexity and accuracy on 9 zero-shot benchmark tasks. All baseline results for QuaRot, SpinQuant, and OSTQuant are reproduced using their official open-source implementations, with necessary modifications to support Qwen models, which differ from LLaMA primarily through the inclusion of attention bias. Due to the absence of FSDP (Fully Sharded Data Parallel) support in the official OSTQuant repository, its results are limited to models with up to 14B parameters. Complete results are provided in Section B.

| W4A4KV4 | Qwen-2.5 3B | | Qwen-2.5 14B | | Qwen-2.5 32B | | LLaMA-3.1 8B | | LLaMA-3.1 70B | | LLaMA-3.2 1B | |
|---|---|---|---|---|---|---|---|---|---|---|---|---|
| | 0-shot[9] Avg.($\uparrow$) | Wiki ($\downarrow$) | 0-shot[9] Avg.($\uparrow$) | Wiki ($\downarrow$) | 0-shot[9] Avg.($\uparrow$) | Wiki ($\downarrow$) | 0-shot[9] Avg.($\uparrow$) | Wiki ($\downarrow$) | 0-shot[9] Avg.($\uparrow$) | Wiki ($\downarrow$) | 0-shot[9] Avg.($\uparrow$) | Wiki ($\downarrow$) |
| Full-Precision | 64.17 | 8.03 | 70.95 | 5.29 | 71.11 | 5.02 | 68.70 | 6.23 | 73.76 | 2.81 | 55.89 | 9.75 |
| QuaRot | 44.30 | 69.33 | 67.23 | 6.77 | 68.14 | 6.04 | 63.74 | 7.82 | 69.56 | 5.31 | 48.66 | 14.44 |
| SpinQuant | 46.86 | 46.35 | 67.29 | 6.55 | 68.51 | 5.88 | 64.58 | 7.51 | 70.69 | 4.74 | 49.41 | 13.46 |
| OSTQuant | 50.81 | 20.09 | 67.81 | 6.37 | OOM | OOM | 64.91 | 7.40 | OOM | OOM | **50.85** | 12.84 |
| **BASE-Q** | **58.93** | **10.43** | **69.17** | **6.28** | **70.18** | **5.65** | **65.36** | **7.17** | **71.54** | **4.17** | 50.61 | **12.66** |
| W4A4KV4 | Llama-2 7B | | Llama-2 13B | | Llama-2 70B | | LLaMA-3 8B | | LLaMA-3 70B | | LLaMA-3.2 3B | |
| | 0-shot[9] Avg.($\uparrow$) | Wiki ($\downarrow$) | 0-shot[9] Avg.($\uparrow$) | Wiki ($\downarrow$) | 0-shot[9] Avg.($\uparrow$) | Wiki ($\downarrow$) | 0-shot[9] Avg.($\uparrow$) | Wiki ($\downarrow$) | 0-shot[9] Avg.($\uparrow$) | Wiki ($\downarrow$) | 0-shot[9] Avg.($\uparrow$) | Wiki ($\downarrow$) |
| Full-Precision | 65.22 | 5.47 | 67.62 | 4.88 | 71.57 | 3.32 | 68.11 | 6.14 | 73.82 | 2.86 | 63.59 | 7.81 |
| QuaRot | 61.59 | 6.12 | 65.03 | 5.39 | 70.28 | 3.76 | 62.89 | 7.82 | 68.76 | 5.62 | 55.86 | 10.07 |
| DFRot | 61.80 | 6.25 | 64.95 | 5.43 | 68.78 | 4.02 | 62.94 | 7.91 | 69.62 | 5.03 | - | - |
| SpinQuant | 61.38 | 5.99 | 65.63 | 5.30 | 70.22 | 3.71 | 63.80 | 7.49 | 69.93 | 5.11 | 57.74 | 9.32 |
| OSTQuant | 62.08 | 5.92 | 65.43 | 5.24 | OOM | OOM | 64.72 | 7.36 | OOM | OOM | 59.29 | 9.16 |
| **BASE-Q** | **62.50** | **5.85** | **65.95** | **5.19** | **70.74** | **3.59** | **65.60** | **7.12** | **71.83** | **4.06** | **60.02** | **9.01** |

Table 2: Summary of MMLU average accuracy (5-shot) comparison. BASE-Q demonstrates consistent performance gains, especially on challenging models like Qwen2.5-3B.

| Method | Llama-2-7B | Llama-2-13B | Llama-3-8B | Qwen2.5-3B | Qwen2.5-14B |
|---|---|---|---|---|---|
| FP16 | 45.87 | 55.23 | 65.30 | 65.70 | 79.73 |
| QuaRot | 39.28 | 50.10 | 55.96 | 28.68 | 72.67 |
| SpinQuant | **40.69** | 50.05 | 56.25 | 29.92 | 73.15 |
| **BASE-Q** | 40.65 | **51.57** | **57.82** | **53.80** | **75.49** |

we fuse the online bias and scaling operations introduced by BASE-Q with quantization and de-quantization into a single Triton kernel (Figure 6), rendering the additional costs of bias and scaling computation negligible. For query and key projections (with small dimensions, e.g., 128×128), we implement the Hadamard transformation directly as a matrix multiplication. For the much larger *down* layer, we apply a single Cooley-Tukey step, ensuring that all compute-intensive kernels operate on manageable, high-parallelism sub-matrices. As shown in Figure 5, our approach yields $2.1\times$ to $2.4\times$ acceleration across all batch sizes compared to standard FP16. Notably, our optimizations incur only minimal overhead versus pure INT4 quantization without any online operations. During the decoding phase, which is typically characterized by a small batch size, the limited parallelism makes it challenging to gain throughput improvements from standard INT4 GEMM kernels. We found that a more effective strategy in this memory-bandwidth-bound scenario is to perform an INT4-FP16 GEMV (Wang et al., 2024) kernel, which yields a significant 1.8x to 2.3x throughput

Table 3: Performance comparison for W3A4KV4 quantization. PPL is measured on WikiText2, and Avg.acc represents the average accuracy on Zero-shot Common Sense Reasoning tasks.

| Method | Qwen2.5-3B | | Qwen2.5-14B | | Llama2-7B | | Llama2-13B | | Llama3-8B | |
|---|---|---|---|---|---|---|---|---|---|---|
| | PPL($\downarrow$) | Avg.acc($\uparrow$) | PPL($\downarrow$) | Avg.acc($\uparrow$) | PPL($\downarrow$) | Avg.acc($\uparrow$) | PPL($\downarrow$) | Avg.acc($\uparrow$) | PPL($\downarrow$) | Avg.acc($\uparrow$) |
| FP16 | 8.03 | 65.80 | 5.29 | 72.68 | 5.47 | 66.70 | 4.88 | 69.98 | 6.14 | 69.97 |
| Quarot | 89.33 | 45.81 | 7.33 | 67.70 | 6.81 | 59.50 | 5.87 | 64.02 | 9.87 | 61.37 |
| Spinquant | 56.36 | 47.40 | 7.15 | 68.47 | 6.72 | 60.13 | 5.78 | 64.83 | 8.70 | 61.61 |
| Flatquant | 10045 | 30.62 | 6.95 | **69.94** | 6.48 | 61.49 | 5.52 | **66.60** | 8.49 | 63.03 |
| **Base-Q** | **11.77** | **56.26** | **6.83** | 67.35 | **6.21** | **61.71** | **5.45** | 65.55 | **7.955** | **63.98** |

increase. Furthermore, across all decoding scenarios, our method consistently achieves a 3.4x to 3.6x reduction in memory footprint through quantized weights and KV-cache(Table 6).

## 5.3 ABLATION STUDY

We perform a systematic ablation study on 3B to 8B models to assess the effects of different quantization strategies in BASE-Q, alongside comparisons with Quarot, SpinQuant, and OSTQuant. Our experimental results, presented in Table 4 indicate that bias correction yields substantial perplexity reductions for Qwen2.5-3B and Llama3-8B, highlighting the importance of addressing inter-channel bias in these models. In contrast, its effect on Llama2-7B remains marginal. Notably, asymmetric scaling delivers consistent improvements across all three models. A detailed analysis of bias correction is provided in Section C.

Table 4: Ablation Study on WikiText2 (word perplexity$\downarrow$)

| Method | Fixed Rotation | Learned Rotation | Bias Corect. | Asym. Scale | Scale | Qwen2.5-3B | Llama-2-7B | Llama-3-8B |
|---|---|---|---|---|---|---|---|---|
| QuaRot | $R_{res}$ $R_v$ $R_{qk}$ $R_{down}$ | | | | | 69.33 | 6.12 | 7.82 |
| | $R_{res}$ $R_{qk}$ $R_{down}$ | $R_v$ | | | | 54.38 -5.95 | 6.18 +0.06 | 7.63 -0.19 |
| | $R_{res}$ $R_{qk}$ $R_{down}$ | $R_v$ | ✓ | | | 13.59 -40.79 | 6.12 -0.06 | 7.42 -0.21 |
| | $R_{res}$ $R_{qk}$ $R_{down}$ | $R_v$ | ✓ | ✓ | | 10.82 -2.77 | 5.92 -0.20 | 7.22 -0.20 |
| **BASE-Q** | $R_{res}$ $R_{qk}$ $R_{down}$ | $R_v$ | ✓ | ✓ | ✓ | 10.83 +0.01 | 5.85 -0.07 | 7.14 -0.08 |
| SpinQuant | $R_{qk}$ $R_{down}$ | $R_{res}$ $R_v$ | | | | 46.35 | 5.99 | 7.49 |
| OSTQuant | $R_{qk}$ $R_{down}$ | $R_{res}$ $R_v$ | | | ✓ | 20.09 | 5.92 | 7.36 |

We conducted additional ablation studies with global rotations using three distinct approaches: standard Hadamard matrices, random Hadamard matrices, and learnable rotations (following SpinQuant's official codebase). In SpinQuant (cf. Fig. 4 in (Liu et al., 2024)), the authors observed that using random Hadamard rotations could cause large fluctuations and subpar quantization performance. Based on this observation, SpinQuant advocates for learnable global rotations to stabilize and improve quantization results under their framework.

However, according to Table 5, quantization metrics showed negligible differences between different rotation types within our BASE-Q framework across five benchmark LLMs. This suggests that any potential improvements from learnable global rotations are effectively subsumed by BASE-Q's bias correction component. Therefore, the additional memory and computational cost introduced by learnable global rotations become unnecessary for maintaining quantization quality in our method.

Table 5: Ablation study on $R_{res}$ choice.

| W4A4KV4 | $R_{res}$ **Setting** | Qwen-2.5 3B | | Qwen-2.5 14B | | LLaMA-2 7B | | LLaMA-3 8B | | LLaMA-2 13B | |
|---|---|---|---|---|---|---|---|---|---|---|---|
| | | 0-shot[9] Avg.($\uparrow$) | Wiki ($\downarrow$) | 0-shot[9] Avg.($\uparrow$) | Wiki ($\downarrow$) | 0-shot[9] Avg.($\uparrow$) | Wiki ($\downarrow$) | 0-shot[9] Avg.($\uparrow$) | Wiki ($\downarrow$) | 0-shot[9] Avg.($\uparrow$) | Wiki ($\downarrow$) |
| Full-Precision | | 64.17 | 8.03 | 70.95 | 5.29 | 65.22 | 5.47 | 68.11 | 6.14 | 67.62 | 4.88 |
| QuaRot | Random Hadamard | 44.30 | 69.33 | 67.23 | 6.77 | 61.59 | 6.12 | 62.89 | 7.82 | 65.03 | 5.39 |
| SpinQuant | Learned Rotation | 46.86 | 46.35 | 67.29 | 6.55 | 61.38 | 5.99 | 63.80 | 7.49 | 65.63 | 5.30 |
| **BASE-Q** | Standard Hadamard | **58.93** | **10.43** | **69.17** | **6.28** | **62.08** | **5.85** | **65.33** | **7.13** | **65.89** | **5.20** |
| | Random Hadamard | **58.60** | **10.43** | **69.22** | **6.28** | **62.24** | **5.87** | **65.21** | **7.14** | **65.46** | **5.20** |
| | Learned Hadamard | **57.95** | **10.44** | **68.43** | **6.26** | **62.68** | **5.86** | **65.01** | **7.13** | **65.60** | **5.20** |

## 6 CONCLUSION

In this work, we analyze key challenges in rotation-based quantization for large language models, identifying channel mean discrepancies and cumulative clipping-induced energy loss as major sources of quantization errors. To address these issues, we propose BASE-Q, which introduces blockwise bias correction and per-channel asymmetric scaling to achieve accurate and efficient quantization with minimal resource overhead. Comprehensive experiments on established LLM benchmarks demonstrate that BASE-Q narrows the accuracy gap to floating-point baselines while substantially reducing memory usage, enabling single-GPU quantization even for large-scale models. Our results highlight BASE-Q as a practical and scalable approach to quantizing LLMs.

REPRODUCIBILITY STATEMENT

To ensure the reproducibility of our results, we have included comprehensive details of our methodology, experimental setup, and all hyperparameters in the main paper and its appendices. We will release our source code and quantized model checkpoints to facilitate verification and future work. An anonymized version of the code and checkpoints will be made available during the rebuttal period, and a public release will follow upon acceptance of the paper.

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

## A  SPEED-UP WITH KERNEL FUSION

We implement kernel fusion to merge the bias, scaling, quantization, and dequantization operations into a single computational step, which reduces on-chip memory access. For the online Hadamard transform, we find that the fast algorithm (Dao, 2023) adopted in Quarot, which uses the recursive Cooley-Tukey method to achieve $O(nlogn)$ complexity, does not fully leverage tensor core parallelism on modern GPUs. To address this, we implement the Hadamard transform as a matrix multiplication in Triton, better utilizing available compute. For the prefill stage, we accelerate int4 GEMM using the CUTLASS library. In the decoding stage, especially with low batch sizes, we leverage the BitBLAS (Wang et al., 2024) library to accelerate int4-fp16 GEMV computations.

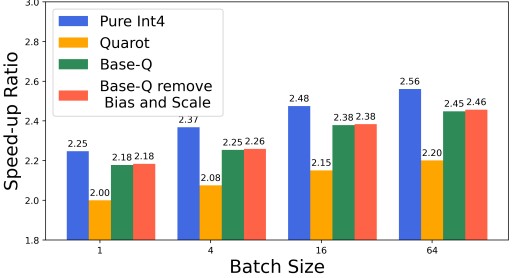

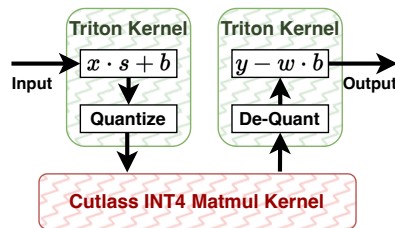

Figure 5: Prefill speedup for Llama2-7B with seqlens 2048 as batch size scales from 1 to 64.

Figure 6: Illustration of kernel fusion.

Table 6: Memory saving on decoding stage for Llama-2-7B.

| bsz× seqlens | 1× 256 | 1× 1024 | 1× 4096 | 16× 256 | 16×1024 | 16×4096 |
|---|---|---|---|---|---|---|
| Fp16 | 12.82 | 13.13 | 14.74 | 14.71 | 20.82 | 45.26 |
| W4A4KV4 | 3.70 | 3.81 | 4.25 | 4.26 | 5.88 | 12.44 |
| Saving Ratio | 71.1% | 71.0% | 71.2% | 71.0% | 71.8% | 72.5% |

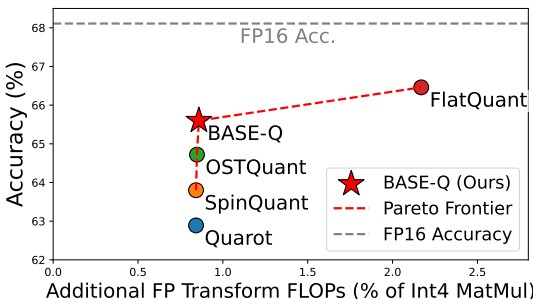

Figure 7: Pareto frontier plot between accuracy and inference cost.

## B  FULL QUANTIZATION RESULTS

We present comprehensive quantization results in this section, including perplexity on WikiText2 and zero-shot accuracy on nine evaluation datasets. All baseline results, including QuaRot, SpinQuant, and OSTQuant, are reproduced using their official open-source implementations, with necessary modifications to support the Qwen model, which differs from LLaMA mainly by including attention bias. As the official OSTQuant repository does not support FSDP (Fully Sharded Data Parallel) training, its results are limited to models with up to 14B parameters, while larger models, such as the 70B model, encounter out-of-memory (OOM) issues. Results for the Llama2 series are summarized in Table 7, those for the Llama3 series are in Table 8, and for the Qwen2.5 series in Table 9. We also perform additional bit-widths (W6A6, W3A4, W2A4) to better demonstrate the robustness of BASE-Q.

Table 7: Complete comparison of the perplexity score on WikiText2 and accuracy on Zero-shot Common Sense Reasoning tasks for **Llama-2 models**.

| Model | #Bits W-A-KV | Method | ARC-c (↑) | ARC-e (↑) | BoolQ (↑) | HellaS. (↑) | Lam. (↑) | OBQA (↑) | PIQA (↑) | SIQA (↑) | WinoG. (↑) | Avg. (↑) | Wiki2 (↓) |
|---|---|---|---|---|---|---|---|---|---|---|---|---|---|
| 2-7B | 16-16-16 | Full Precision | 46.33 | 74.54 | 77.74 | 76.02 | 73.90 | 44.20 | 79.05 | 46.16 | 69.06 | 65.22 | 5.47 |
| | 4-4-4 | Quarot | 42.15 | 70.37 | 73.00 | 73.09 | 70.68 | 39.40 | 77.26 | 43.30 | 65.04 | 61.59 | 6.12 |
| | | SpinQuant | 40.78 | 70.45 | 73.79 | 72.40 | 71.30 | 38.40 | 75.63 | 43.45 | 66.22 | 61.38 | 5.99 |
| | | OSTQuant | 41.64 | 68.94 | 74.43 | 73.17 | 71.61 | **42.20** | 77.09 | **43.71** | 65.90 | 62.08 | 5.92 |
| | | **BASE-Q** | **42.24** | **71.42** | **74.74** | **73.69** | 71.32 | 41.80 | **77.53** | 43.35 | **66.38** | **62.50** | **5.85** |
| 2-13B | 16-16-16 | Full Precision | 49.15 | 77.44 | 80.61 | 79.38 | 76.73 | 45.20 | 80.52 | 47.39 | 72.14 | 67.62 | 4.88 |
| | 4-4-4 | Quarot | 46.76 | 75.08 | 76.97 | 75.84 | 74.36 | 42.80 | 78.84 | 45.19 | 69.46 | 65.03 | 5.39 |
| | | SpinQuant | **48.46** | 74.33 | 77.28 | 76.27 | 74.83 | 44.60 | 78.67 | **46.01** | 70.24 | 65.63 | 5.30 |
| | | OSTQuant | 47.10 | 75.20 | 77.46 | **77.71** | **75.14** | 44.60 | 78.67 | 45.75 | 68.03 | 65.41 | 5.24 |
| | | **BASE-Q** | 47.01 | **75.67** | **78.90** | 77.42 | 74.87 | 44.60 | **79.27** | 45.34 | **70.48** | **65.95** | **5.19** |
| 2-70B | 16-16-16 | Full Precision | 57.25 | 81.06 | 83.76 | 83.82 | 79.62 | 48.80 | 82.70 | 49.18 | 77.98 | 71.57 | 3.32 |
| | 4-4-4 | Quarot | 55.97 | **80.18** | 81.87 | 82.25 | 78.73 | 48.00 | 81.39 | 47.49 | 75.69 | 70.28 | 3.76 |
| | | SpinQuant | 54.78 | 79.76 | 81.90 | 82.78 | 79.20 | 47.40 | 81.77 | 48.46 | **76.80** | 70.22 | 3.71 |
| | | OSTQuant | OOM | OOM | OOM | OOM | OOM | OOM | OOM | OOM | OOM | OOM | OOM |
| | | **BASE-Q** | **56.40** | 80.09 | **82.17** | **82.79** | **79.37** | **48.20** | **82.32** | **48.62** | 76.72 | **70.74** | **3.59** |

Table 8: Complete comparison of the perplexity score on WikiText2 and accuracy on Zero-shot Common Sense Reasoning tasks for **Llama-3 models**.

| Model | #Bits W-A-KV | Method | ARC-c (↑) | ARC-e (↑) | BoolQ (↑) | HellaS. (↑) | Lam. (↑) | OBQA (↑) | PIQA (↑) | SIQA (↑) | WinoG. (↑) | Avg. (↑) | Wiki2 (↓) |
|---|---|---|---|---|---|---|---|---|---|---|---|---|---|
| 3-8B | 16-16-16 | Full Precision | 53.33 | 77.74 | 81.35 | 79.15 | 76.01 | 45.00 | 80.79 | 47.13 | 72.53 | 68.11 | 6.14 |
| | 4-4-4 | Quarot | 46.08 | 70.50 | 74.50 | 74.47 | 70.58 | 40.60 | 76.50 | 44.93 | 67.88 | 62.89 | 7.82 |
| | | SpinQuant | 47.35 | 73.36 | 75.75 | 74.74 | 70.70 | 41.20 | 77.04 | 44.93 | 69.14 | 63.80 | 7.49 |
| | | OSTQuant | 48.21 | 72.69 | **79.02** | 75.69 | 70.52 | **44.00** | 77.86 | 44.98 | 69.53 | 64.72 | 7.36 |
| | | **BASE-Q** | **50.51** | **76.05** | 78.26 | **76.48** | **71.22** | 43.60 | **78.89** | **45.75** | **69.61** | **65.60** | **7.12** |
| 3-70B | 16-16-16 | Full Precision | 64.33 | 85.90 | 85.23 | 84.89 | 79.82 | 48.60 | 84.55 | 50.72 | 80.35 | 73.82 | 2.86 |
| | 4-4-4 | Quarot | 53.16 | 78.11 | 82.91 | 80.80 | 75.49 | 44.40 | 79.65 | 47.49 | 76.87 | 68.76 | 5.62 |
| | | SpinQuant | 57.25 | 80.22 | 83.06 | 81.86 | 75.04 | 46.20 | 81.88 | 47.44 | 77.27 | 69.93 | 5.11 |
| | | OSTQuant | OOM | OOM | OOM | OOM | OOM | OOM | OOM | OOM | OOM | OOM | OOM |
| | | **BASE-Q** | **59.98** | **82.70** | **84.65** | **83.92** | **78.42** | **47.20** | **83.08** | **48.41** | **78.14** | **71.83** | **4.06** |
| 3.1-8B | 16-16-16 | Full Precision | 53.50 | 81.19 | 82.08 | 78.90 | 75.82 | 44.80 | 81.23 | 47.19 | 73.56 | 68.70 | 6.23 |
| | 4-4-4 | Quarot | 46.08 | 72.31 | 77.34 | 74.46 | 71.36 | 42.40 | 76.82 | 44.37 | 68.51 | 63.74 | 7.82 |
| | | SpinQuant | 47.87 | 76.05 | 76.76 | 74.63 | 70.54 | **43.20** | **79.27** | 45.70 | 67.17 | 64.58 | 7.51 |
| | | OSTQuant | 47.44 | 75.34 | **78.84** | 75.48 | **71.38** | 41.80 | 78.24 | 47.13 | 68.51 | 64.91 | 7.40 |
| | | **BASE-Q** | **49.15** | **77.19** | 78.56 | **76.41** | 70.72 | 42.40 | 79.16 | **45.85** | **69.22** | **65.36** | **7.17** |
| 3.1-70B | 16-16-16 | Full Precision | 64.93 | 86.66 | 85.41 | 85.02 | 79.16 | 48.00 | 84.28 | 50.56 | 79.79 | 73.76 | 2.81 |
| | 4-4-4 | Quarot | 57.51 | 80.35 | 83.27 | 81.45 | 75.65 | 44.80 | 81.66 | 45.60 | 75.77 | 69.56 | 5.31 |
| | | SpinQuant | 59.13 | 82.28 | **84.37** | 82.24 | 76.32 | 46.00 | 82.21 | 47.24 | 76.40 | 70.69 | 4.74 |
| | | OSTQuant | OOM | OOM | OOM | OOM | OOM | OOM | OOM | OOM | OOM | OOM | OOM |
| | | **BASE-Q** | **60.49** | **84.13** | 83.12 | **83.10** | **77.06** | **47.40** | **83.24** | **47.85** | **77.43** | **71.54** | **4.17** |
| 3.2-1B | 16-16-16 | Full Precision | 36.35 | 60.48 | 64.07 | 63.65 | 62.93 | 37.20 | 74.54 | 42.99 | 60.77 | 55.89 | 9.75 |
| | 4-4-4 | Quarot | 31.23 | 51.01 | 59.42 | 54.74 | 43.90 | 34.40 | 66.92 | 40.48 | 55.88 | 48.66 | 14.44 |
| | | SpinQuant | 32.68 | 51.68 | 58.47 | **56.68** | 47.22 | **34.80** | 67.79 | 40.23 | 55.17 | 49.41 | 13.46 |
| | | OSTQuant | **33.45** | **55.26** | **61.80** | 56.02 | **48.90** | 33.80 | 70.02 | **41.61** | 56.83 | **50.85** | 12.84 |
| | | **BASE-Q** | 31.23 | 53.91 | 59.79 | 56.61 | 46.73 | 31.60 | **70.57** | 40.74 | **56.91** | 49.79 | **12.63** |
| 3.2-3B | 16-16-16 | Full Precision | 45.90 | 71.63 | 73.33 | 73.60 | 70.48 | 43.00 | 77.58 | 46.98 | 69.85 | 63.59 | 7.81 |
| | 4-4-4 | Quarot | 38.40 | 59.13 | 64.56 | 66.49 | 60.00 | 37.20 | 72.25 | 42.68 | 62.04 | 55.86 | 10.07 |
| | | SpinQuant | 37.12 | 60.90 | 68.72 | 68.83 | 61.67 | 39.60 | 73.88 | 44.58 | 64.40 | 57.74 | 9.32 |
| | | OSTQuant | 41.04 | **68.06** | 68.84 | 68.75 | **63.50** | 40.40 | 74.21 | **44.78** | 64.01 | 59.29 | 9.16 |
| | | **BASE-Q** | **41.64** | 66.86 | **72.45** | **69.83** | 62.57 | **40.80** | **75.84** | 44.52 | **65.67** | **60.02** | **9.01** |

Table 9: Complete comparison of the perplexity score on WikiText2 and accuracy on Zero-shot Common Sense Reasoning tasks for **Qwen2.5 Models**.

| Model | #Bits W-A-KV | Method | ARC-c (↑) | ARC-e (↑) | BoolQ (↑) | HellaS. (↑) | Lam. (↑) | OBQA (↑) | PIQA (↑) | SIQA (↑) | WinoG. (↑) | Avg. (↑) | Wiki2 (↓) |
|---|---|---|---|---|---|---|---|---|---|---|---|---|---|
| 2.5-3B | 16-16-16 | Full Precision | 47.53 | 73.06 | 77.22 | 73.52 | 67.11 | 42.00 | 78.84 | 49.80 | 64.14 | 64.17 | 8.03 |
| | 4-4-4 | Quarot | 30.72 | 52.69 | 51.16 | 49.52 | 20.63 | 34.60 | 66.70 | 38.84 | 53.83 | 44.30 | 69.33 |
| | | SpinQuant | 34.47 | 57.58 | 54.04 | 50.91 | 24.14 | 32.20 | 68.01 | 40.48 | 57.93 | 46.86 | 46.35 |
| | | OSTQuant | 38.48 | 58.08 | 61.90 | 56.19 | 36.33 | 37.60 | 67.90 | 42.48 | 58.33 | 50.81 | 20.09 |
| | | **BASE-Q** | **42.75** | **67.26** | **71.96** | **66.21** | **52.24** | **38.40** | **74.59** | **46.01** | **63.77** | **58.13** | **10.83** |
| 2.5-14B | 16-16-16 | Full Precision | 58.87 | 79.17 | 85.26 | 82.91 | 74.62 | 45.20 | 82.05 | 55.17 | 75.30 | 70.95 | 5.29 |
| | 4-4-4 | Quarot | 55.80 | 79.46 | 80.24 | 78.40 | 70.06 | 41.60 | 78.45 | 50.20 | 70.88 | 67.23 | 6.77 |
| | | SpinQuant | 53.75 | 79.97 | 78.87 | 79.18 | **70.97** | 44.00 | 79.22 | 49.23 | 70.40 | 67.29 | 6.55 |
| | | OSTQuant | 54.78 | 78.91 | 80.73 | **79.81** | 69.18 | **44.40** | 79.71 | 50.15 | 72.61 | 67.81 | 6.37 |
| | | **BASE-Q** | **56.14** | **82.41** | **82.23** | 79.75 | 70.89 | 42.80 | **80.20** | **52.97** | **72.69** | **68.90** | **6.28** |
| 2.5-32B | 16-16-16 | Full Precision | 55.63 | 80.93 | 87.19 | 84.06 | 76.96 | 44.00 | 82.32 | 56.29 | 75.22 | 71.11 | 5.02 |
| | 4-4-4 | Quarot | 52.05 | 74.66 | 85.41 | 81.70 | 73.14 | 42.40 | 79.98 | 52.81 | 71.11 | 68.14 | 6.04 |
| | | SpinQuant | 52.82 | 76.05 | 85.11 | 80.99 | 72.71 | 44.40 | 80.03 | 52.15 | 72.30 | 68.51 | 5.88 |
| | | OSTQuant | OOM | OOM | OOM | OOM | OOM | OOM | OOM | OOM | OOM | OOM | OOM |
| | | **BASE-Q** | **54.61** | **78.16** | **87.06** | **82.49** | **75.61** | **44.60** | **81.12** | **53.63** | **74.35** | **70.18** | **5.65** |

Table 10: Performance comparison for W6A6KV6 quantization. PPL is measured on WikiText2, and Avg.acc represents the average accuracy on Zero-shot Common Sense Reasoning tasks.

| Method | Qwen2.5-3B | | Qwen2.5-14B | | Llama2-7B | | Llama2-13B | | Llama3-8B | |
|---|---|---|---|---|---|---|---|---|---|---|
| | PPL(↓) | Avg.acc(↑) | PPL(↓) | Avg.acc(↑) | PPL(↓) | Avg.acc(↑) | PPL(↓) | Avg.acc(↑) | PPL(↓) | Avg.acc(↑) |
| FP16 | 8.03 | 65.80 | 5.29 | 72.68 | 5.47 | 66.70 | 4.88 | 69.98 | 6.14 | 69.97 |
| Quarot | 8.13 | **66.09** | 5.39 | 72.53 | 5.64 | 65.22 | **4.90** | 69.29 | 6.23 | 69.78 |
| Spinquant | 8.13 | 65.82 | 5.39 | 72.54 | 5.50 | 66.58 | **4.90** | 69.11 | 6.23 | **69.92** |
| RoLoRa | – | – | – | – | – | **67.10** | – | 68.80 | – | 68.10 |
| Omniquant | – | – | – | – | 7.48 | 58.65 | 6.74 | 61.02 | – | – |
| **Base-Q** | **8.12** | 65.45 | **5.38** | **72.59** | **5.49** | 66.68 | **4.90** | 69.40 | **6.21** | 69.48 |

Table 11: Performance comparison for W2A4 quantization. PPL is measured on WikiText2, and Avg.acc represents the average accuracy on Zero-shot Common Sense Reasoning tasks.

| Method | Qwen2.5-3B | | Qwen2.5-14B | | Llama2-7B | | Llama2-13B | | Llama3-8B | |
|---|---|---|---|---|---|---|---|---|---|---|
| | PPL(↓) | Avg.acc(↑) | PPL(↓) | Avg.acc(↑) | PPL(↓) | Avg.acc(↑) | PPL(↓) | Avg.acc(↑) | PPL(↓) | Avg.acc(↑) |
| FP16 | 8.03 | 65.80 | 5.29 | 72.68 | 5.47 | 66.70 | 4.88 | 69.98 | 6.14 | 69.97 |
| Quarot | 549.48 | 36.64 | 18.74 | 49.96 | 65.22 | 39.14 | 16.58 | 43.09 | 73.08 | 38.78 |
| Spinquant | 430.54 | 36.62 | **17.81** | **52.29** | 60.82 | 37.13 | 15.56 | 44.32 | 61.68 | 40.57 |
| FlatQuant | NaN | NaN | NaN | NaN | NaN | NaN | NaN | NaN | NaN | NaN |
| **Base-Q** | **22.86** | **44.09** | 49.78 | 47.13 | **14.01** | **48.62** | **14.20** | **57.95** | **21.92** | **45.93** |

## C    THE ABLATION STUDY OF BIAS CORRECTION

To clarify the effectiveness of bias correction, we analyzed the behavior of bias terms during quantization by examining the cosine similarity between the initialized channel means and the optimized bias correction coefficients on Qwen2.5-3B. Except for the first block—where channel mean deviation is not yet present, as shown in Figure 2 and table 12—the optimized bias values in all subsequent layers remain highly correlated with the initial channel means (cosine similarity >97.5%), which fully aligns with our theoretical analysis.

Table 12: Cosine similarity between blocks after quantization.

| Block index | 0 | 1 | 2 | 3 | 4 | 5 | 6 | 7 |
|---|---|---|---|---|---|---|---|---|
| Cosine Similarity | 0.556 | 0.976 | 0.979 | 0.997 | 0.988 | 0.988 | 0.988 | 0.989 |

# D VISUALIZATION OF ACTIVATION DISTRIBUTION

We provide visualization results for the input activations of the multi-head attention blocks and MLP blocks at the 1st, 11th, and 31st layers of Qwen2.5-3B, Llama2-7B, and Llama3-8B. For each selected layer, we illustrate the distributions of activations under various rotation strategies. We observe that inter-channel mean misalignment consistently occurs in different layers and across all evaluated models, and that learned rotations are insufficient to eliminate this issue. As discussed in Section 3.2, this misalignment is a significant source of quantization error, which can be effectively mitigated by our proposed bias correction technique.

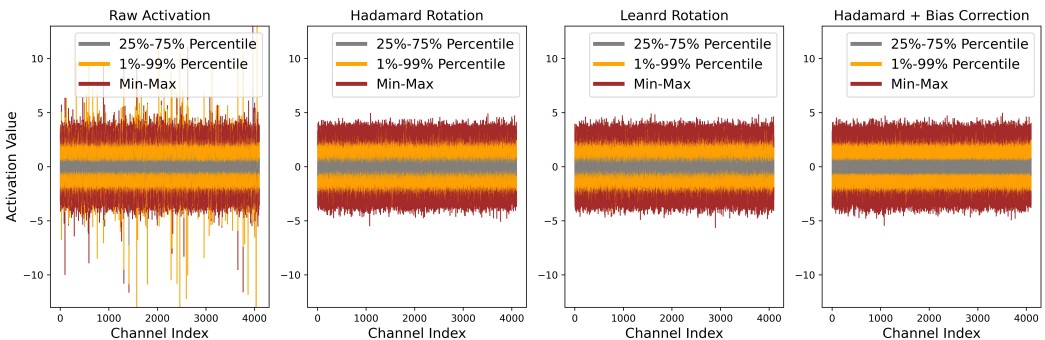

Figure 8: Visualizations comparing the activation distributions from the 1st MHSA block in Llama2-7B.

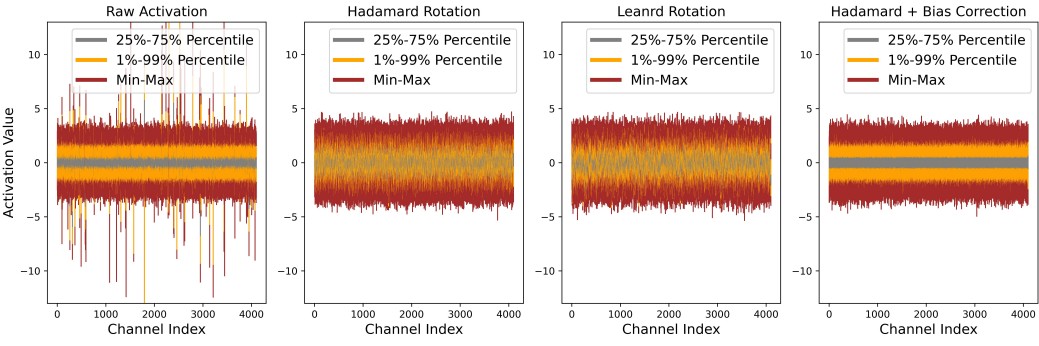

Figure 9: Visualizations comparing the activation distributions from the 1st MLP block in Llama2-7B.

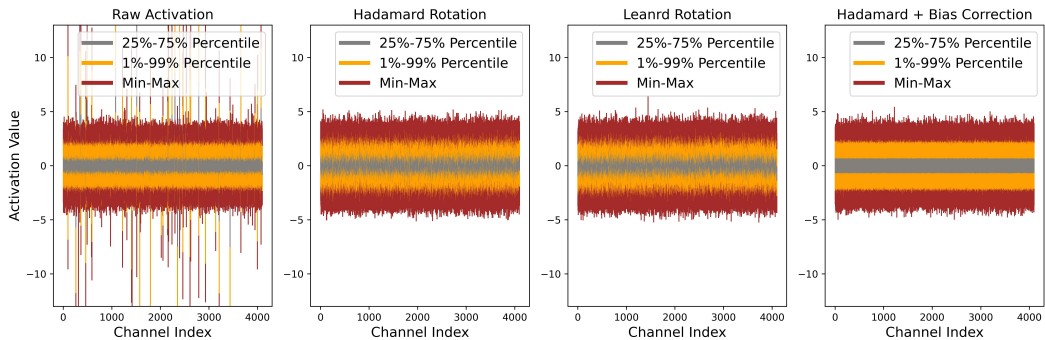

Figure 10: Visualizations comparing the activation distributions from the 11th MHSA block in Llama2-7B.

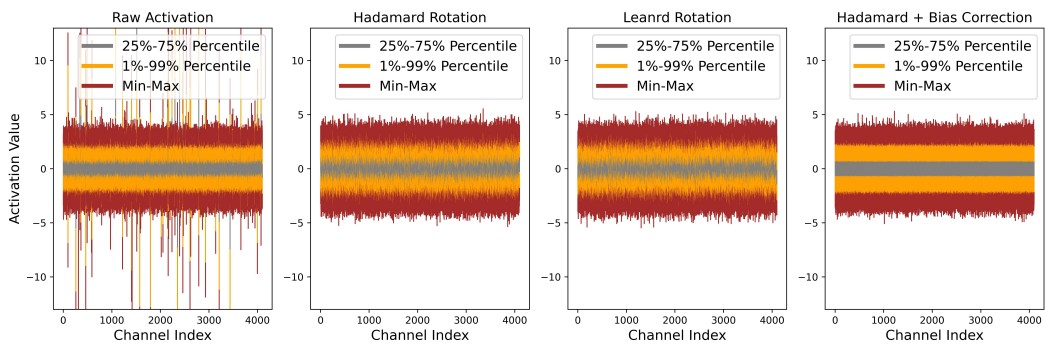

Figure 11: Visualizations comparing the activation distributions from the 11th MLP block in Llama2-7B.

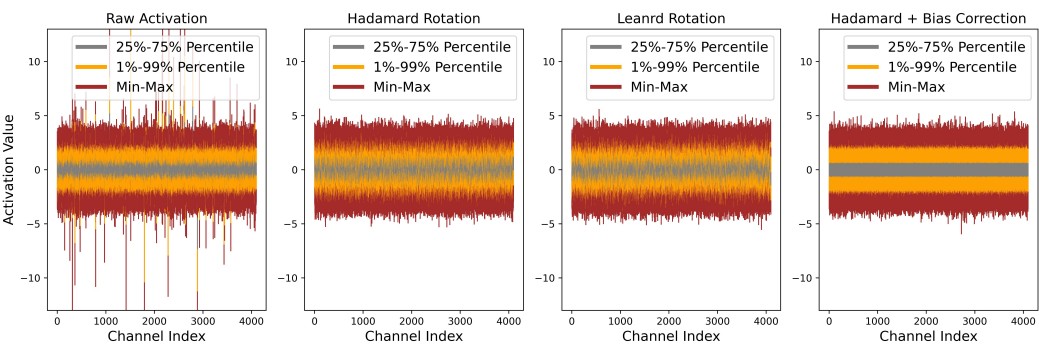

Figure 12: Visualizations comparing the activation distributions from the 31st MHSA block in Llama2-7B.

# E  THE USE OF LARGE LANGUAGE MODELS (LLMS)

This paper was partially created with the assistance of a Large Language Model (LLM), which was used for tasks such as sentence polishing, brainstorming, and content organization. All content has been finally reviewed and confirmed by the author.

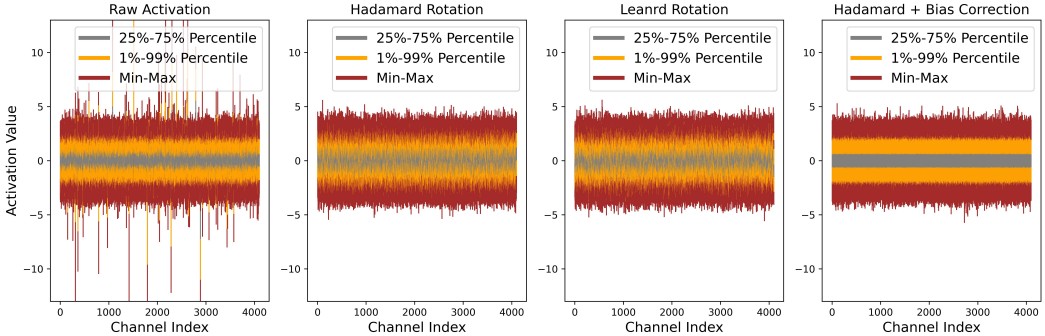

Figure 13: Visualizations comparing the activation distributions from the 31st MLP block in Llama2-7B.

