# OpenReview forum: "BASE-Q: Bias and Asymmetric Scaling Enhanced Rotational Quantization for Large Language Models"
_ICLR.cc/2026/Conference — Submitted to ICLR 2026_

### Official Review · Reviewer_gCv9 · 2025-10-22

**Soundness:** 3
**Presentation:** 3
**Contribution:** 3
**Rating:** 8
**Confidence:** 5

**Summary:**

This paper proposes BASE-Q, a lightweight 4-bit post-training quantization framework for large language models (LLMs) that enhances rotational quantization through two key components: blockwise bias correction to eliminate channel-mean misalignment and asymmetric scaling to compensate for clipping-induced energy loss. The authors argue that instead of pursuing costly global rotation learning (e.g., SpinQuant, OSTQuant), it is more effective to directly correct the residual quantization errors that persist even after optimal rotations. The method is evaluated on a wide range of modern LLMs—including Llama-2/3/3.1/3.2 and Qwen2.5 series with strong empirical results.

**Strengths:**

1) The shift—from “optimize rotation” to “correct residual errors”—is conceptually fresh and represents one of the few recent works that rethinks the purpose of rotation in quantization, rather than merely tuning it.

2) The paper has a relatively rigorous theoretical analysis on the error decomposition of rotational quantization.

3) The evaluated LLMs are relatively recent.

4) The corresponding GPU kernels are implemented, and the practical speedups of quantized LLMs using their method are justified.

**Weaknesses:**

1) The proposed blockwise bias correction introduces an additive term $b_c$ before activation quantization and compensates it post-multiplication (Eq. 14). If I remember correctly, some linear layers in modern LLMs (e.g., attention projections) do not include a learnable bias in their original architecture. It is unclear whether the introduced correction term incurs additional latency or memory overhead during inference in such bias-free layers.

2) In Equation (1), it seems the variable $e_i$  is used without definition.

3) The paper argues that global orthogonal rotations cannot eliminate channel-mean variance. However, it does not discuss whether locally (layerwise) learned invertible transformations—such as those in FlatQuant—could mitigate this issue. If such methods can reduce channel-mean misalignment, how does BASE-Q’s explicit bias correction compare in efficacy or efficiency?

4) Could non-orthogonal but invertible transformations simultaneously reduce both channel-wise variance and inter-channel mean discrepancy?

5) Figure 6 illustrates a fused pipeline combining Triton and CUTLASS kernels. How are memory layouts, synchronization, and data movement coordinated between the Triton-based quantization/dequantization and the CUTLASS-based INT4 GEMM? Are these kernels truly fused—or are their speeds measured separately and summed for total throughput?

6) The paper states that 70B-model quantization takes ~10 hours on a single A800. Could authors break down the time spent on joint optimization of $R_v, b_c, s_a$, and $α$? What fraction of this time is attributable to the proposed bias and asymmetric scaling components versus baseline rotation and scaling?

7) The MSE objective in Equation (7) appears to lack an explicit scale factor in the reconstruction term.

9) Eq.(12) gives an optimal global scaling $s^2 = \mathbb{E}[\|w\|^2] / \mathbb{E}[\|a\|^2]$ from error propagation (Eq.11).
Yet BASE-Q uses per-quantizer asymmetric $s_a$ and blockwise symmetric $s_j$, not $s$.
Clarifying the link—or empirically validating $s_a$'s alignment with this principle—would strengthen the argument.

**Questions:**

See weaknesses.

---

> ### Author Response · Authors · 2025-11-22
>
> We are immensely grateful for your exceptionally positive and insightful review. We are particularly encouraged by your recognition of our core conceptual contribution—shifting focus from 'optimizing rotation' to 'correcting residual errors'—and your appreciation for our theoretical analysis and practical speedups. Your detailed questions are highly pertinent, and addressing them has significantly strengthened our paper.
>
> ---------------------
> **W1: On the Inference Overhead of Bias Correction**
>
> Thank you for raising this important practical concern. You are correct that naively adding a bias term to bias-free layers in PyTorch would incur latency. However, we have addressed this through efficient kernel engineering.
>
> 1. **Kernel Fusion:** The bias subtraction (  $b_c$ ) and the de-quantization process are both element-wise operations. As illustrated in **Figure 6** of our paper, we **fuse these two operations into a single custom Triton kernel**.
> 2. **Latency Analysis:** This fusion allows for efficient data reuse within the GPU's registers and shared memory, effectively hiding the latency of the bias operation. Our profiling shows that the primary overhead for such operations comes from kernel launch time, not the low-complexity element-wise computations themselves. As a result, the fused kernel exhibits **no observable latency increase** compared to a de-quantization-only kernel.
>
> Therefore, our bias correction introduces negligible parameter overhead and, thanks to kernel fusion, **zero additional inference latency**, even in layers that were originally bias-free.
>
> -------------------------
> **W2: Clarification of the Notation $e_i$**
>
> Thank you for your careful reading and for pointing out this omission. In the revised manuscript, we have now clarified in the text following Equation (1) that $e_i$ represents the i-th standard basis vector (a one-hot vector), which is marked in blue.
>
> -----------------------------
> **W3: Discussion on Layerwise Transforms (e.g., FlatQuant)**
>
> This is an excellent question that gets to the heart of different PTQ philosophies. You are right that layerwise invertible transforms, like in FlatQuant, could also potentially reduce channel-mean variance. We conducted new experiments to compare their efficacy and efficiency with our explicit bias correction.
>
> **1. Efficacy and Optimization Potential:**
> We applied our bias correction **on top of** the FlatQuant framework. Interestingly, our results showed **no further accuracy improvement**. This suggests that FlatQuant's dense, layer-specific online transformations are already so powerful that they exhaust most of the optimization potential within each layer, including implicitly correcting the channel-mean misalignment.
> |Method|Wiki-2-ppl|
> |--------------------------|---------|
> |FlatQuant|5.79|
> |FlatQuant+Bias Cor. + Asy.Scale|5.79|
>
> **2. Efficiency and Deployment Trade-off:**
> However, the key difference lies in efficiency. FlatQuant achieves this at the cost of inducing significant online computation overhead, as shown in our Pareto frontier plot (** Figure 7 in revised manuscript**). In contrast, our method leverages a fusible global rotation and a lightweight, fused bias correction, achieving competitive accuracy with substantially lower overhead.
>
> In conclusion, while layerwise transforms can also address channel-mean variance, our explicit bias correction achieves a similar goal with vastly superior efficiency and deployment-friendliness. We have added a discussion of this comparison to Section 5.4 in our paper.
>
> -----------------
> **W4: On the Efficacy of Non-Orthogonal Transformations**
>
> This reveals a core trade-off in designing efficient quantization methods:
>
> Layerwise Invertible Transforms (e.g., FlatQuant): These methods can apply both rotation and fine-grained scaling transformations on a per-layer basis. This allows them to not only eliminate channel variance and mean discrepancies but also suppress the propagation of quantization error (as discussed in Section 3.4). However, this flexibility comes at the high cost of non-fusible, online computations for every layer.
> BASE-Q's Approach: We prioritize efficiency by using a fusible global rotation, which incurs zero overhead and can smooth most key layers (Q, K, V, O, Up, Gate) simultaneously. This choice intentionally forgoes the ability to apply per-layer scaling within the rotation framework but compensates by introducing a highly efficient, blockwise bias correction to specifically target the residual mean discrepancy.
>
> Table: Percentage of remaining inter-channel mean discrepancy after different transformations.
>
> | Transformation Type      | Block 1 | Block 2 | Block 3 | Block 4 |
> |--------------------------|---------|---------|---------|---------|
> | Global Hadamard Rotation | 0.49    | 0.41    | 0.32    | 0.34    |
> | Layerwise Non-Orthogonal | 0.05    | 0.03    | 0.05    | 0.04    |
> | Layerwise Orthogonal     | 0.05    | 0.05    | 0.06    | 0.05    |

---

> ### Author Response · Authors · 2025-11-22
>
> ----------------------------------------
> **W5: Clarification on Kernel Fusion in Figure 6**
>
> Thank you for the detailed question. To clarify, the Triton and CUTLASS kernels operate independently and are not fused together.
>
> The 'fusion' illustrated in Figure 6 refers to a **Triton-only fusion**: we fused our proposed bias correction with the quantization/de-quantization steps into a single Triton kernel. This fused Triton kernel is then called separately from the CUTLASS kernel, which is solely responsible for the INT4 GEMM computation.
>
> The latency measurements reported are for the **entire end-to-end pipeline**, not a sum of separately measured kernels.
>
> -------------------
> **W6: Breakdown of Blockwise Optimization Time**
>
> Thank you for the question. For a single block of a 70B model, the optimization time is broken down as follows: \~2.5 minutes for $R_v$ and $s$, \~2.5 minutes for GPTQ weight quantization, and \~4 minutes for optimizing our proposed bias, asymmetric scaling components and clipping factors ( $b_c$ $s_a$ and $alpha$).
>
>
> --------------
>
> **W7: On the MSE Objective in Equation (7)**
>
> Thank you for the question. The MSE objective in Equation (7) is defined to measure the error between the **original, full-precision input $x$** and its **reconstructed, de-quantized output**.
>
> The full expression, implicitly represented in Equation (7), is:
> $E[ || x - (DeQuant(Quant(x))) ||² ]$
> where $DeQuant(Quant(x))$ expands to $F_clip(F_{round}((x-θ)/(...)), ...) * s + θ$. Our objective is thus to find the optimal clipping threshold $θ$ (which determines the scale $s$) that minimizes the error against the original $x$.
>
> The term $x-θ$ inside the $F_{round}$ function is part of the quantization process itself (shifting the input before scaling and rounding), not the reconstruction target. We have revised the text surrounding Equation (7) to make this distinction clearer (marked in **blue**).
>
> --------------
> **W8: On the Intention of Eq. (12)**
>
> Thank you for this insightful question. We apologize for not making the intention behind Section 3.4 and Equation (12) clearer.
>
> Our goal with Eq. (12) was to illustrate a key principle: for a linear layer, applying a symmetric scaling factor (s) can provably reduce error propagation. This principle is orthogonal to rotation, meaning they can be used together to minimize the quantization error of a linear layer.
>
> However, a critical challenge arises when applying this to fusible rotational quantization. The validity of fusing a global rotation relies on the fact that rotation preserves the per-token activation energy established by RMSNorm. Symmetric scaling, unfortunately, breaks this per-token energy invariance, making it incompatible with fusible rotations for layers immediately following an RMSNorm (i.e., $Q$,$ K$,$ V$, $Up$, $Gate$ projections). It can only be effectively paired with fusible rotation in layers like the $O$ and $Down$ projections.
>
> This analysis serves two purposes:
>
> It justifies the effectiveness of combining symmetric scaling and rotation where possible (e.g., in our $O$ and $Down$ layers).
>
> It reveals, from first principles, why methods using layerwise online invertible transforms (like FlatQuant) can theoretically achieve higher accuracy. They can apply both symmetric scaling and rotation to every layer, at the cost of losing fusibility.
> Therefore, rotational quantization methods enjoy the efficiency of fusion but inherently sacrifice the accuracy gains that symmetric scaling could bring to the $Q$, $K$, $V$, $Up$, and $Gate$ layers. We have revised the text surrounding Eq. (12) to explicitly state this crucial trade-off and clarify our intention.
>
> To empirically validate the effectiveness of the principle outlined in Eq. (12), we conducted an ablation study on the symmetric scaling factors for the O and Down layers, which are compatible with fusible rotation. We compared three distinct settings: 1) No Scaling; 2) Ones Initialization and then fine-tuned; 3) Eq. (12) Initialization: The scaling factors are initialized with the theoretical optimal values derived from Eq. (12) and then fine-tuned.
>
> The results, shown below for the first four blocks of Llama-2-7B, demonstrate two key findings. First, initializing with the theoretical values from Eq. (12) consistently yields the lowest final MSE loss. Second, the high cosine similarity between the final learned scales and their initial theoretical values indicates that our practical optimization converges to a solution that remains in close alignment with the theoretical principle.
>
> ||Block 0 |Block 1 |Block 2 |Block 3 |
> |---|--|----|----|----|
> |MSE loss with out sym-scaling|3.0e-6|4.7e-4|8.3e-5|1.5e-4|
> |MSE loss with ones initialization|2.8e-6|4.6e-4|8.0e-5|1.4e-4|
> |MSE loss with Eq.12 initialization|2.6e-6|4.4e-4|7.6e-5|1.3e-4|
> |Cosine Similarity (scaling of O_proj)|0.976| 0.985|0.924|0.971|

---

### Official Review · Reviewer_cUMP · 2025-10-31

**Soundness:** 2
**Presentation:** 2
**Contribution:** 2
**Rating:** 4
**Confidence:** 5

**Summary:**

This paper proposes BASE-Q, a lightweight quantization framework for LLMs. BASE-Q addresses two key error sources in rotational quantization—channel mean misalignment and clipping-induced energy loss—through blockwise bias correction and asymmetric scaling, avoiding the high cost of optimizing global rotation matrices.

**Strengths:**

1. Compared with the previous approach of optimizing the global rotation matrix, this method can be optimized with fewer GPU resources.
2. The authors conducted extensive experiments on both QWen series and LLaMA series models under the W4A4KV4 quantization configuration. Across these models, BASE-Q consistently achieved performance improvements.

**Weaknesses:**

1. The author only tested the effectiveness of the proposed method in basic experiments such as Zero-Shot and PPL evaluations. How does this method perform on more complex benchmark datasets like MMLU?
2. While I acknowledge the rationality of the authors' method, it appears that their approach is essentially OminiQuant under rotation conditions. Additionally, the fused-bias technique they employed is not particularly novel—Outlier Suppression++ has also adopted the fused-bias technology. Therefore, the authors' technical contributions are somewhat limited.
3. How long does the authors' method take to quantize a model? The comparison of quantization time remains an important aspect.
4. How is the generalization ability of the authors' method, and does the selection of the calibration dataset affect the performance of the algorithm?
5. Beyond INT4 quantization, does the authors' method exhibit good performance under other quantization modes such as NVFP4?
6. The authors have omitted some important works: Xiang, J. and Zhang, S.Q., 2024. DFRot: Achieving Outlier-Free and Massive Activation-Free for Rotated LLMs with Refined Rotation. arXiv preprint arXiv:2412.00648.

**Questions:**

See weekness.

---

> ### Author Response · Authors · 2025-11-22
>
> We appreciate your thorough evaluation and recognition of BASE-Q's key advantages in resource efficiency and empirical performance. To address your concerns about evaluation breadth and technical novelty, we have provided substantial new experiments and clarifications below that demonstrate our work's robustness and unique contributions.
>
> ---------------
> **Q1: Additional Evaluation on the MMLU Benchmark**
>
> Following your recommendation, we have conducted new MMLU (5-shot) evaluations on five representative models. The results, presented below, show that BASE-Q consistently maintains its superiority over prior methods, confirming that its benefits extend to complex reasoning tasks. This new table has been added to the main experimental section of our revised manuscript for a comprehensive comparison.
>
>
> | model     | Llama-2-7B | Llama-2-13B | Llama-3-8B | Qwen2.5-3B | Qwen2.5-14B |
> | --------- | ---------- | ----------- | ---------- | ---------- | ----------- |
> | FP16      | 45.87      | 55.23       | 65.30      | 65.70      | 79.73       |
> | Quarot    | 39.28      | 50.10       | 55.96      | 28.68      | 72.67       |
> | Spinquant | **40.69**  | 50.05       | 56.25      | 29.92      | 73.15       |
> | BASE-Q    | 40.65      | **51.57**   | **57.82**  | **53.80**  | **75.49**   |
>
>
> ---------------
> **W2: Clarifying the Novelty of BASE-Q**
>
> Thank you for your critique. While the techniques we use (bias correction, asymmetric scaling) are not new, our primary contribution is **theoretical, not methodological**.
>
> Specifically, our work is the first to:
>
> 1. **Systematically analyze and identify the key residual errors** in rotational quantization: rounding errors from channel-mean variance and clipping-induced energy loss (Section 3).
> 2. **Demonstrate that correcting these specific errors makes expensive global rotation optimization unnecessary** (Table 3), challenging a core assumption from prior work like SpinQuant.
>
> Therefore, the bias correction and asymmetric scaling in our work should be viewed as the simplest, most direct tools to validate our theoretical findings. Our novelty lies in this new understanding, which enables a more targeted and efficient quantization framework.
>
> ---------------
> **W3: Quantization Time and Resource Efficiency**
>
> Our method is highly resource-efficient. We can quantize a 70B model in approximately 10 hours on a single NVIDIA A800 GPU.
>
> In sharp contrast, methods requiring full-model optimization, such as SpinQuant, demand significantly more resources—requiring 5 A800 GPUs and a total of 30 GPU-hours for a similar task. This stark difference highlights the practical advantage of our blockwise approach, rendering advanced quantization accessible without large-scale hardware clusters.
>
> We have added these timing and resource comparisons to the Deployment Details (Section 5) in our revised manuscript, which is marked in blue.

---

> ### Author Response · Authors · 2025-11-22
>
> -------------------------------
> **W4:Robustness to Calibration Dataset Selection**
>
> Our approach, grounded in mathematical equivalence principles, is designed to be robust to the choice of calibration data. To validate this, we conducted two new experiments on the Llama-2-7B model.
>
> **1. Effect of Calibration Set Size:**
> We varied the size of the Wikitext-2 calibration set from 64 to 512 samples. As shown in the table below, performance saturates quickly; increasing the size beyond 128 samples yields minimal gains. This indicates that our method can effectively learn the necessary quantization parameters from a small, accessible dataset.
>
> | Calibration Size | WikiText2 PPL (↓) | Zero-shot Avg. Acc. (↑) |
> | :--------------- | :---------------- | :---------------------- |
> | 64               | 5.91              | 62.31                   |
> | 128              | 5.87              | 62.50                   |
> | 256              | 5.85              | 62.50                   |
> | 512              | 5.86              | 62.52                   |
>
> **2. Effect of Calibration Set Source:**
> We replaced the in-domain Wikitext-2 dataset with an out-of-domain dataset, RedPajama. The results show only a negligible impact. While perplexity in Wikitext-2 slightly increased, the zero-shot accuracy remained almost identical, and our method maintained its performance advantage over baselines such as SpinQuant.
>
> | Calibration Source (Samples) | WikiText2 PPL (↓) | Zero-shot Avg. Acc. (↑) |
> | :--------------------------- | :---------------- | :---------------------- |
> | Wikitext-2 (256)             | 5.85              | 62.50                   |
> | RedPajama (256)              | 5.90              | 62.54                   |
>
>
> -------------------------------
> **W5: Performance Analysis with FP4 Quantization**
>
> Thank you for this excellent question regarding emerging formats like NVFP4.
>
> To investigate this, we tested our framework on Llama-2-7B by replacing the INT4 activator quantizer with an NVFP4 quantizer, while keeping weight quantization as INT4 (as our GPTQ-based weight quantization currently does not support non-uniform formats). The results show a slight performance degradation compared to our standard INT4 activation setting.
>
> | Activation Format | WikiText2 PPL (↓) | Zero-shot Avg. Acc. (↑) |
> | :---------------- | :---------------- | :---------------------- |
> | **INT4**          | **5.85**          | **62.50**               |
> | NVFP4             | 5.90              | 62.41                   |
>
> Our analysis attributes this to the interaction between the post-rotation distribution and the quantizer's design.
>
> 1. **Post-Rotation Distribution:** Our method and previous rotational quantization methods, through rotation, transforms the activation distribution to be more compact and **Gaussian**-like.
> 2. **Quantizer Properties:** NVFP4 excels at representing heavy-tailed distributions due to its wide dynamic range. However, for a compact Gaussian distribution, its 15 non-uniform levels may be less efficient than the 16 uniformly spaced levels of INT4.
>
> To verify this, we directly measured the Mean Squared Error (MSE) of quantizing a standard Gaussian signal with both formats, using an optimally searched scale. The results below confirm that for a Gaussian source, INT4 provides a slightly better representation.
>
> |    **Gaussian**     | **INT4**   | **NVFP4 (FP4)** |
> | :------ | :--------- | :-------------- |
> | **MSE** | **0.0128** | 0.0130          |
>
> In summary, while FP4 is highly effective for un-rotated activations with prominent outliers, rotation's "Gaussianization" effect makes the uniform INT4 quantizer a more natural and slightly more effective fit. This finding highlights an interesting trade-off between data transformation techniques and quantization formats.
>
> -------------------------------
> **Q6: Comparison with DFRot**
>
> Thank you for bringing this recent paper to our attention. We have now added a comparison with DFRot. Our experimental results on models including Llama-2-7/13/70B and Llama-3-8/70B show that **BASE-Q consistently outperforms DFRot**. We have **updated the main comparison tables and Rlated work** in the revised manuscript.

---

### Official Review · Reviewer_fJor · 2025-11-02

**Soundness:** 3
**Presentation:** 2
**Contribution:** 3
**Rating:** 4
**Confidence:** 4

**Summary:**

BASE-Q is a lightweight post-training quantization framework for large language models that achieves near-full-precision performance at 4-bit weight, activation and KV-cache (W4A4KV4) quantization. BASE-Q combines fixed Hadamard rotations with two block-wise corrections—channel-wise bias to cancel rounding error caused by misaligned channel means and asymmetric scaling. Extensive results show that BASE-Q narrows the accuracy gap to FP16 by 50.5%, 42.9% and 29.2% vs. QuaRot, SpinQuant and OSTQuant across 12 open-source LLMs (1B–70B) on WikiText-2 perplexity and nine zero-shot tasks, while fused Triton kernels deliver 2.1–2.4× prefill speed-up and 71% memory reduction. While the paper is well motivated by math analysis, the methodology is kind of incremental compared with previous methods, and there are still issues with the presentation (as listed in weakness and detailed questions).

**Strengths:**

- The authors conduct extensive empirical results to demonstrate the effectiveness of BASE-Q against a set of leading baselines such as SpinQuant and OSTQuant. The ablation study in Table 2 also looks adequate.

- The authors also customize the kernel to further speed up the inference of BASE-Q. Figure 5 looks promising, and the proposed method introduces little overhead.

**Weaknesses:**

- The proposed method is largely built upon existing frameworks like SpinQuant. Both bias correction and asymmetric scaling are kind of incremental to the baseline. Moreover, the training paradigm (e.g., blockwise training) also follows SpinQuant.

- The writing is kind of hard to follow. Some equations should be explained in more detail (e.g., Equation 7, 8). The necessary derivations are missing. The logic can be a bit messy. For instance, it can be hard to understand the expectation of rounding error (Line 155), yet this is only introduced in more detail later in Equation 10.

- While I appreciate the analysis in Section 3, it seems to overlap quite well with the proposed BASE-Q in Section 4. This makes the reading back-and-forth.

**Questions:**

- FlatQuant should be considered for comparisons in the main table, since both the model and quantization settings are pretty aligned with its original paper. The results of FlatQuant in Table 9 and Table 10 look weird.

- Section 3.4 is hard to follow. It seems weird why ϵ follows uniform distribution in Equation 9 and Normal distribution in Equation 10. Besides, I doubt the assumption that ϵ follows the uniform distribution, as the rounding error also depends on the distribution of weights, which can be normally distributed.

- It is not clear how the blockwise training is conducted, e..g, what is the definition of a block?

---

> ### Author Response · Authors · 2025-11-22
>
> We greatly appreciate your insightful summary and positive feedback, particularly regarding our empirical results and kernel speedups. Your constructive comments have been addressed in detail below to clarify our novelty and improve the presentation.
>
> -------------------------
> **W1: Distinction from SpinQuant**
>
> Thank you for the comment. We respectfully disagree that our method is an incremental extension of SpinQuant. The core difference lies in our optimization paradigm.
>
> SpinQuant requires expensive, end-to-end training across the full model to optimize a single global rotation matrix ($R_{res}$). In contrast, our method, BASE-Q, utilizes a **fixed** global rotation (e.g., Hadamard) and introduces lightweight, **blockwise** corrections for bias and scaling. This distinction is fundamental. By processing only one block at a time, we can quantize a 70B model on a single 40GB GPU, whereas SpinQuant requires at least five 80GB GPUs for the same task.
>
> Furthermore, our ablation studies (Table 3) demonstrate that combining our method with a globally *learned* rotation (as in SpinQuant) yields no significant improvement over using a simple, fixed Hadamard matrix. This suggests that our blockwise corrections more fundamentally address the residual errors post-rotation, rendering the costly global rotation optimization unnecessary.
>
> Therefore, our approach not only improves accuracy and drastically reduces optimization costs, but more importantly offers a **scheme change** for rotational quantization that provides a more targeted and efficient solution to the performance limitations.
>
>
> -----------------------
> **W2: Improving Clarity and Logical Flow in Explanations and Equations**
>
> Thank you for your feedback on our paper's clarity. We agree that some explanations were not detailed enough and the logical flow could be improved.
>
> In the revised manuscript, we have provided more detailed explanations for critical equations and concepts. Specifically:
>
> - We have expanded the text surrounding **Equations 7 and 8** to better explain the derivation of the optimal clipping threshold and the resulting energy loss.
> - We have addressed the logical flow issue you pointed out regarding the rounding error. The explanation on **Line 155** is now better connected to the detailed formulation in Equation 10, ensuring concepts are introduced in a more sequential manner.
>
> All new and revised texts are marked in blue for your convenience. We believe these changes have significantly improved the readability of the paper.
>
> -------------------------
> **W3: Improving the Connection Between Problem Analysis (Sec. 3) and Solution (Sec. 4)**
>
> Thank you for your feedback on the paper's structure. We design our flow by first analyzing and identifying the true limitation of rotational quantization in Section 3 and then utilize our observation to design our method in Section 4. We agree that the connection between the analysis in Section 3 and our method in Section 4 can be made more explicit to improve the reading flow.
>
> To address this, we have added a concluding paragraph at the end of Section 3. This new paragraph summarizes the key problems identified in our analysis and explicitly signposts how they are solved by the corresponding components of our method presented in Section 4. This bridges the two sections, aleviating the need to read back and forth.

---

> ### Author Response · Authors · 2025-11-22
>
> -------------------
> **Q1: Comparison with FlatQuant**
>
> Thank you for the valuable suggestion. We agree that a comparison with FlatQuant is important.
>
> Our initial decision to place it in the appendix was due to our paper's focus on optimizing rotational quantization, whereas FlatQuant introduces additional non-rotational online transforms. Given the page limits, we prioritized direct rotational baselines in the main text. We have now moved this comparison to the main text (Section 5.4) and expanded our analysis as follows.
>
> Performance at W4A4: The Pareto frontier of accuracy versus overhead (Figure 7) perfectly illustrates the trade-off at W4A4. FlatQuant achieves a higher accuracy, but at the cost of significant computational overhead. In contrast, BASE-Q remains on the Pareto frontier, offering a superior balance for practical deployment.
>
> Performance at Lower Bit-widths (W3A4/W2A4): At these aggressive settings, the "**weird**" results you correctly noted become apparent. This is not an evaluation error but a crucial finding about FlatQuant's practical robustness. It suffers from numerical stability issues, leading to model collapse (e.g., on Qwen2.5-3B at W3A4) or failure to converge (at W2A4). Our method, guided by a clearer theoretical objective and the stability of fixed transforms, proves far more robust.
>
> Underlying Principle: This performance difference stems from a fundamental trade-off. As discussed in Section 3.4, combining scaling with rotation is key to minimizing error. FlatQuant's layerwise online transforms can apply both to every layer, explaining its high potential accuracy. However, fusible rotational methods face a constraint: scaling breaks the per-token energy invariance required for fusion with layers following an RMSNorm (Q, K, V, Up, Gate). This means rotational methods inherently sacrifice some theoretical accuracy on these layers in exchange for zero overhead, a trade-off that BASE-Q navigates effectively.
>
> In summary, while FlatQuant pushes for maximum accuracy at a high and sometimes unstable cost, BASE-Q provides a robust, efficient, and deployment-friendly solution that remains on the Pareto frontier.
>
> -------------------
> **Q2: Clarification on the Quantization Noise Model in Section 3.4**
>
> Thank you for these sharp questions regarding Section 3.4. We acknowledge that the transition from a uniform to a normal distribution was not explained sufficiently. Let us clarify the logic.
>
> Why Uniform Distribution (Eq. 9)? Our analysis in this section focuses on the rounding error of activations post-rotation. For a multi-level uniform quantizer, it is a standard and well-established model to approximate the rounding error as an additive noise uniformly distributed between ±$\Delta$/2, where $\Delta$ is the quantization step size. This holds when $\Delta$ is small relative to the signal's dynamic range. This is the noise model in the rotated space.
>
> Why Normal Distribution (Eq. 10)? The transition to a Normal distribution in Eq. 10 arises from viewing this error's effect from the perspective of the original, pre-rotation space.
>
> The uniform rounding error ϵ from Eq. 9 is a high-dimensional vector where each component is an independent random variable.
>
> To analyze its impact in the original space, we must apply the inverse rotation (R⁻¹) to this noise vector.
> According to the Central Limit Theorem's principles in high dimensions, a linear combination of many independent random variables (which is what each element of the rotated-back noise vector is) tends towards a Gaussian distribution. Therefore, the effect of the uniform rounding noise from the rotated domain is equivalent to adding isotropic Gaussian noise in the original domain.
>
> The orthogonal rotation preserves the L2-norm, so the total variance remains unchanged, i.e., $\Delta$²/12.
>
> In the revised manuscript, we have substantially rewritten Section 3.4 to make this step-by-step reasoning explicit, clarifying the distinction between the noise model in the rotated space versus the original space, and better explaining the role of the rotation.
>
> -----------------------
> **Q4: Clarification on the Definition of a 'Block'**
>
> Thank you for the question. A 'block' in our paper refers to a **Transformer block** (i.e., a self-attention module followed by an MLP). Our blockwise training means we optimize the quantization parameters for one such Transformer block at a time.
>
> We have now explicitly added this definition in the Introduction and Methodology sections to ensure clarity.

---

> ### Comment · Reviewer_fJor · 2025-11-27
>
> I appreciate the authors' response and explanations, which paritially address my concerns. I still have concerns w.r.t. the technical contribution. While I agree there are technical differences from SpinQuant, the overall idea to employ rotational matrices or blockwise training have been actually explored by quite a few works (e.g, QuaRot, OmniQuant, FlatQuant). The writing and logic flow can be still improved. Given these factors I would choose to keep my score.

---

> > ### Author Response · Authors · 2025-11-27
> > **BASE-Q understands and resolves residual errors left behind by existing work on optimizing rotation**
> >
> > Thank you for your comments. We are glad that your concerns are resolved.
> >
> > We would like to point out your misunderstanding on our contributions. As stated in our introduction, in the paragraph of line 67, this work shifts away from the recently prevailing research trend of further optimizing rotation. Instead, we make fundamental analysis on the source of residual quantization errors and provide theory-guided techniques to resolve such errors. In this way, the contribution of our paper is **NOT** simply proposing another rotation or block-wise optimization method, but to expose and resolve an overlooked source of quantization error that can enable both more effective and more efficient calibration in LLM quantization. This would guide future research away from the misleading assumption that "an optimal global rotation is the key to minimizing quantization error"
> >
> > We are also pleased that our contributions are well-recognized by all other reviewers. Reviewer gCv9 specificlly mentions **"The shift—from “optimize rotation” to “correct residual errors”—is conceptually fresh and represents one of the few recent works that rethinks the purpose of rotation in quantization, rather than merely tuning it."** Reviewer i5oe believes **"The paper explains the reason behind all of the design choices in the proposed method."** And Reviewer cUMP identifies that our method **"avoiding the high cost of optimizing global rotation matrices."** All these comments support the contribution of this work.

---

### Official Review · Reviewer_i5oe · 2025-11-04

**Soundness:** 3
**Presentation:** 1
**Contribution:** 3
**Rating:** 6
**Confidence:** 3

**Summary:**

The paper suggests that while Hadamard rotations (or any globally learned rotation matrix) help with outliers, they fail to fully reduct the variance between channel means to zero which results in poor quantization. To alleviate this, the authors suggest using a learnable bias correction term. Combined with a fixed scaling (offline) and using assymetric quantization, the authors name the obtained method BASE-Q. The results show better performance than several existing methods (QuaRot, SpinQuant, and OSTQuant) after quantization over different architectures including LLaMA 2, 3 and QWEN.

**Strengths:**

The paper explains the reason behind all of the design choices in the proposed method. The method is effective and shows strong results compared with existing SoTA methods that are used as baselines. The authors include ablations to showcase the importance of each component.

In addition to theoretical efficiency arguments, the authors implemented optimized kernels that allow them to obtain real-world speedup more than 2x when using 4-bit quantization.

**Weaknesses:**

The writing can be improved quite a lot. Important details are either missing or are scattered in different places. To name a few exampels:

1. The dimensions of newly introduced parameters are not clearly specified. It is hard to understand whether a parameter is scalar, a vector or a matrix.

2. There are two different scaling mentioned in the paper. These are referred by different names in different places. For example, Table 2 calls them unpaired scale and scale. There is no other mentioned of "Unpaired" anywhere else in the text and the latter scale is usually referred to as "Dynamic Assymetric Scale".

As another note, using assymetric quantization can be combined with any of the existing methods which makes it an orthogonal axis. Similarly, all the low-level optimizations to handle the assymetric scale such as fused kernels can also be re-used for existing methods. While it makes sense to use the best recipe to obtain the final results (i.e. the best possible performance of quantized models), it adds an additional confounder when comparing different methods. Still, this issue is mostly resolved by Table 2 which shows the main effect comes from the main contributions of the method.

**Questions:**

1. Is there an $s^{-1}$ missing from Eq. 14 to be applied on the activations?  Similarly shouldn't $s^{a}$ be inverted before multiplying by the quantized weights?

2. Just to confirm is $R_{qk}$ shared between layers? (and similarly other learned rotation matrices such as $R_{v}$)

---

> ### Author Response · Authors · 2025-11-22
>
> We sincerely thank you for your positive feedback on our method's effectiveness, strong empirical results, real-world speedup, as well as for your constructive suggestions. We also thank you for the valuable feedback to further improve our paper.
>
> ---------------------------------------------------------
> **W1: Clarification of Parameter Notations:**
>
> We agree that a clear and consistent notational system is crucial for readability.
> Instead of specifying the dimension for every single parameter, which would make the paper verbose, we have adopted a more systematic approach as you rightly suggested. In the revised manuscript, we have added a dedicated 'Notation' paragraph **at the beginning of the Section 3**. This paragraph explicitly defines our convention:
>
> Matrices are denoted by uppercase bold letters (e.g., `$\mathbf{W}$`)..
> Vectors are denoted by lowercase bold letters(e.g., `$\mathbf{x}$`)..
> Scalars are denoted by regular lowercase letters(e.g., `$s$`)..
>
> We now **strictly adhere to this convention throughout the paper**. All revisions are **marked in blue** for your convenience.
>
> ---------------------------------------------------------
> **W2: Unifying Terminology for Assymetric Scaling:**
>
> Thank you for pointing out the inconsistent terminology. We sincerely apologize for the confusion.
>
> Our initial use of 'Unpaired scale' in Table 2 was a practical choice for brevity, as the more descriptive term 'Asymmetric Scale' was too long for the table's formatting. However, we failed to provide a clear definition for 'Unpaired scale'. We have replaced the ambiguous term in Table 2 with **`Asym. Scale`**. This new label serves as a clear and concise abbreviation for 'Asymmetric Scale'.
>
> ---------------------------------------------------------
> **Q1：Clarification on Equation 14 and the Role of Asymmetric Scale ($s_a$):**
>
> Thank you for your sharp observation.
>
> Regarding Equation 14, you are absolutely right. The $s^{-1}$ term was indeed missing due to a typo. We apologize for this error and have added the missing $s^{-1}$ to the activation term in the revised manuscript.
>
> Regarding Equation 15, $s_a$ should not be inverted and applied to the weights. This is by design and one of our core contributions. The role of $s_a$ is to compensate for the accuracy loss caused by activation clipping during quantization. Therefore, it does not function as part of an equivalent transformation pair (forward-inverse). Applying it before the quantizer, rather than fusing it into the subsequent weights, is a deliberate choice to maintain the invariance between the quantized and original weights, thereby preserving generalization.
>
> ---------------------------------------------------------
> **Q2. Clarification on the Scope of Rotation Matrices ($R_{qk}$ and $R_v$):**
>
> Thank you for the question. For **$R_{qk}$**, we use a fixed Hadamard matrix. It is therefore identical across all layers, which minimizes storage overhead. For **$R_v$**, its scope is layer-wise. This is practical as its dimension is comparable to that of an attention head, which allows it to be fused into the weights.

---

### Author Response · Authors · 2025-12-03
**Authors' Final Summary for AC**

Dear Area Chair,

We thank all reviewers for their time and effort in evaluating our manuscript. We appreciate the thoughtful feedback, which has helped us to significantly refine and strengthen our work. We have accordingly revised our paper, with all new additions marked in blue.

We would like to provide a concise summary, focusing on two key aspects: (1) the novelty and significance of our contribution, and (2) the extensive revisions and new experiments we have conducted in response to the reviews.

-----
## On the Novelty and Significance of Our Contribution

We respectfully address the concern raised by **Reviewer** **fJor** regarding our work's novelty. We believe this concern stems from a misunderstanding of our core contribution. As correctly identified by other reviewers, our work does **NOT** simply propose another incremental method.

Our primary contribution is a conceptual shift in the field of rotational quantization. Instead of joining the prevailing trend of "optimizing the global rotation matrix," we are one of the first to ask a more fundamental question: What are the residual error sources that persist even after an optimal rotation?

Our key contributions, as recognized by the majority of the reviewers, are:

* **A "conceptually fresh" rethink (Reviewer gCv9):** We shift the focus from “optimizing rotation” to “correcting residual errors.” Our theoretical analysis (Sec. 3) systematically identifies two previously overlooked error sources: channel-mean variance and clipping-induced energy loss.

* **Theory-driven methodology (Reviewer i5oe):** Our proposed techniques (bias correction and asymmetric scaling) are not an arbitrary combination, but rather direct, simple, and theoretically-guided solutions to the specific problems we identified. As Reviewer i5oe noted, "The paper explains the reason behind all of the design choices."

* **Challenging the necessity of costly global optimization (Reviewer cUMP):** A key implication of our findings is that the expensive, multi-GPU optimization of a single global rotation matrix is often unnecessary. By directly and efficiently targeting the true error sources at a block level, we achieve superior results with a fraction of the resources, "avoiding the high cost of optimizing global rotation matrices" (Reviewer cUMP).

In summary, the novelty of our paper is **NOT** in proposing another block-wise method, but in providing a new theoretical understanding that guides the community away from the costly pursuit of an "optimal" global rotation and towards more efficient, targeted error correction.

-----
##  Comprehensive Revisions and New Experiments

We have diligently addressed every concern raised by all reviewers with substantial revisions and extensive new experiments to validate our claims and improve the paper's clarity. The key additions include:

* **Clarity and Presentation (Reviewers i5oe, fJor):**
  * Added a dedicated "Notation" paragraph and unified all symbols for clarity.
  * Unified all terminology (e.g., "Asym. Scale").
  * Provided detailed explanations and derivations for key equations (Eq. 7, 8) and concepts (rounding error analysis in Sec 3.1 & 3.4).

* **Expanded Comparisons and Ablations (Reviewers fJor, gCv9, cUMP):**
  * New MMLU Benchmark Evaluation: Conducted extensive 5-shot MMLU tests on five models, demonstrating BASE-Q's superior performance on complex reasoning tasks.
  * New Comparison with Concurrent Work (DFRot): Added a direct comparison, showing BASE-Q's consistent outperformance.
  * New In-depth Analysis of FlatQuant: Moved the comparison to the main text, provided a Pareto-frontier analysis (Fig. 7), and added a crucial discussion on its numerical instability at lower bit-widths, turning a "weird result" into a key finding.
  * New Ablation Studies: Added new ablations on calibration dataset choice (size and source), FP4 quantization format, transformation module selection, and symmetric scaling initialization strategies.

* **Practical Details (Reviewers fJor, cUMP):**
  * Provided a detailed breakdown of quantization time and resource consumption, highlighting our method's superior efficiency.
  * Clarified the engineering details of our fused kernels and confirmed zero additional latency for our bias correction.

We are confident that these extensive revisions have thoroughly addressed all raised concerns and have significantly strengthened the paper's contribution, clarity, and empirical validation. We respectfully ask you to consider these points and the overwhelmingly positive feedback from the other reviewers in your final assessment.

---

### Meta-Review · Area_Chair_FdTL · 2026-01-04

**Summary:**

This paper received mixed review ratings. All reviewers acknowledged the strong performance and notable speedups achieved by the proposed rotation-based quantization method. However, they also raised substantial concerns regarding the paper’s contribution and novelty, many writing issues, insufficient technical clarity, and insufficient experimental comparisons.

The rebuttal effectively addressed writing issues, clarified technical details, and provided additional experimental results. Nevertheless, the concerns regarding contribution and novelty remain unresolved. The authors’ arguments are largely centered on comparisons with the method, SpinQuant, and primarily emphasize differences relative to that approach. From a broader perspective of outlier mitigation in LLM quantization, the findings and methodology do not appear new when compared with existing approaches such as QuaRot, OmniQuant, and FlatQuant.
In addition, the authors did not directly address R3’s question regarding a detailed novelty discussion in comparison with OmniQuant with rotation and Suppression++.

As a result, this AC cannot recommend acceptance, and hopes the detailed reviewer comments will help improve this work.

**Reviewer Scores:**

I don't think they will change their scores.

---

### Decision · Program_Chairs · 2026-01-26

Reject